# A Survey on Life-Cycle-Oriented Certificate Management in Industrial Networking Environments

**Julian Göppert** *, **Andreas Walz** and **Axel Sikora**

Institute of Reliable Embedded Systems and Communication Electronics (ivESK),
Offenburg University of Applied Sciences, 77652 Offenburg, Germany; axel.sikora@hs-offenburg.de (A.S.)
* Correspondence: julian.goeppert@hs-offenburg.de; Tel.: +49-781-205-4957

**Abstract:** Driven by the Industry 4.0 paradigm and the resulting demand for connectivity in industrial networking, there is a convergence of formerly isolated operational technology and information technology networks. This convergence leads to attack surfaces on industrial networks. Therefore, a holistic approach of countermeasures is needed to protect against cyber attacks. One element of these countermeasures is the use of certificate-based authentication for industrial components communicating on the field level. This in turn requires the management of certificates, private keys, and trust anchors in the communication endpoints. The work at hand surveys the topic of certificate management in industrial networking environments throughout their life cycle, from manufacturing until their disposal. To the best of the authors' knowledge, there is no work yet that surveys the topic of certificate management in industrial networking environments. The work at hand considers contributions from research papers, industrial communication standards, and contributions that originate from the IT domain. In total, 2042 results from IEEE Xplore, Science Direct, Scopus, and Springer Link were taken into account. After applying inclusion and exclusion criteria and title, abstract, and full-text analysis, 20 contributions from research papers were selected. In addition to the presentation of their key contributions, the work at hand provides a synopsis that compares the overarching aspects. This comprises different proposed entity architectures, certificate management functions, involvement of different stakeholders, and consideration of life cycle stages. Finally, research gaps that are to be filled by further work are identified. While the topic of certificate management has already been addressed by the IT domain, its incorporation into industrial communication standards began significantly later and is still the subject of research work.

**Keywords:** certificate; management; credentialing; public key infrastructure; PKI; deployment; onboarding; security configuration; life cycle; industrial automation; fieldbus; ICS; survey

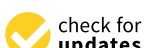



## 1. Introduction

In the past, industrial networks that connected a large number of industrial automation (IA) components like sensors, actuators, or PLCs were separated from information technology (IT) networks by hierarchical gateways [1]. Driven by the Industry 4.0 paradigm and the rapidly increasing demand for connectivity, there is a convergence of formerly separated operational technology (OT) and IT networks [2]. This convergence leads to the adoption of common internet standards in OT networks [3] and to attack surfaces that might be exploited [4]. An overview of the history of exploited vulnerabilities that led to cyber incidents in industrial automation systems is presented by Hemsley et al. [5]. Additionally, a review on cyber vulnerabilities of communication protocols in industrial control systems is presented by Xu et al. [6]. The manipulation of especially safety-critical systems may lead to damage to production equipment, the environment, and people [7].

Therefore, a holistic approach of countermeasures, often titled as a defense-in-depth approach [8], is needed to protect against cyber security incidents. One element of this holistic approach is the implementation of cryptographic protection mechanisms at the

level of IA component communication [8], in the following also referred to as *field-level* communication. A detailed analysis of security objectives and requirements for PROFINET Security, as one representative of an Ethernet-based protocol for field-level communication, is given in a white paper [9]. The use of symmetric cryptography requires the involved endpoints to hold a shared symmetric key for message protection. To proceed, a certificate-based mutually authenticated key agreement, certificates, private keys, and trust anchors are required in the IA components [10]. In order to have certificates, private keys, and trust anchors in IA components, the secure management of these artifacts is required throughout the whole life cycle of an IA component [11]. The required secure and life-cycle-oriented management of certificates motivates the survey at hand. To the best of our knowledge at the time of writing, there is no published work yet that surveys the topic of certificate management in industrial networking environments with regard to the life cycle of an IA component and a focus on field-level communication. Therefore, our work shall contribute the following aspects:

- Identify relevant literature regarding life-cycle-oriented certificate management in industrial networking environments, specifically focusing on field-level communication;
- Present and categorize reviewed works and identify the state of existing research by considering proposals from research papers, industrial communication standards, and IT-domain-based standards;
- Provide a synopsis of the overarching aspects of the selected approaches ;
- Point out research gaps.

The survey follows a methodological-formal approach by first defining research questions, then querying the selected databases IEEE Xplore, Science Direct, Scopus, and Springer Link, defining inclusion and exclusion criteria, screening titles and abstracts, and finally performing a full-text analysis to retrieve information to answer the research questions formulated in Section 3. Starting with 2042 results (without duplicates) taken into account, we reduced this number to 20 research paper contributions. Moreover, we additionally took contributions from industrial communication standards and IT domain standards into account.

The remainder of the paper is structured as follows: Section 2 presents background knowledge, followed by a description of the applied survey methodology in Section 3 to retrace the selection process transparently. Section 4 presents and categorizes work that addresses certificate management. In Section 5, a synopsis of the presented approaches is provided by analyzing and comparing overarching aspects of the individual approaches. Section 6 discusses research gaps before the paper is concluded in Section 7.

## 2. Background

This section introduces background information that will be used in the remainder of the paper. This comprises an explanation of certificates, private keys, and trust anchors as the to-be-managed artifacts and an overview of the acting entities in public key infrastructures (PKIs).

First, this section introduces the artifacts that IA components will be equipped with. This comprises certification paths, private keys, and trust anchors. A *certification path* or *certificate* consists of an EE certificate and potentially one or more sub CA certificates. The EE certificate contains a public key corresponding to a (secret) private key and identity information that identifies the EE instance. The identity information is cryptographically bound to the public key by the EE certificate's signature, which is generated by the issuer of the certificate. In addition to the EE certificate, the intermediate CA certificates may also be contained in the certification path. Among the most widespread representatives of certificates and certification paths is the Internet X.509 Public Key Infrastructure Certificate Profile [12]. However, there also exist other form factors for certificates (e.g., OpenPGP certificates) [13]. Next to the certificate, containing the public key portion, a *private key* or *secret* forms the private portion of an asymmetric key-pair, which shall only be known by the legitimate owner and therefore may be stored in a secure element component. This

secure element component provides persistent storage for keys and a protected execution environment that prevents readout [12]. In order to authenticate to another entity, a subject must send its certification path and prove that it possesses the private key matching the public key in the certificate. The *trust anchor* element is used to initialize the validation of the certification paths of other entities. The trust anchor object may represent some CA and often comes in the form of a certificate [12]. Please note that next to the public key and the identity information, *authorization attributes* may be contained in the certificate. These attributes allow one to validate the permission of an entity's action. In our scope, we are particularly interested in permissions in conjunction with certificate management, that is, if an entity is allowed to manage certificates, private keys, or trust anchors on an IA component. Authorization attributes may be encoded directly in certificates (e.g., by encoding as an extension in an X.509 certificate) [12]. In order to achieve the security goals of authentication, especially mutual authentication, only having certificates in the endpoints is not enough. As explained, this process also incorporates trust anchors and private keys. Please note that, for the sake of simplicity, we sometimes use "certificate management" as an umbrella term. However, please note that in the following, we also consider the management of private keys and trust anchors. The triplet of a certificate, private key, and trust anchor is also referred to as "credential" in the following.

Next, we present the acting entities in a PKI as described in RFC 4210 [14] and RFC 5280 [12], which specify the *Certficate Management Protocol* and the *Internet X.509 PKI Certificate Profile* (see Figure 1). The acting entities comprise end entities, certification authorities, and registration authorities. *End entities* (EEs) are the entities to whom the certificates are issued. In the Internet certificate profile specified by RFC 5280 [12], the identity of an EE is expressed in the *subject* or *subjectAltName* field of a certificate. In the remainder of the paper, end entities that are subject to certificate management processes are IA components. Our understanding of IA components is that they are devices (e.g., sensors, actuators, or PLCs) that exchange data with each other and run one or more applications for the purpose of controlling an automation system. IA components exchange data by using protocols common to industrial networks. They may be (a) Ethernet-based (e.g., PROFINET, Etherned/IP, EtherCAT, or POWERLINK) or (b) non-Ethernet-based (e.g., PROFIBUS DP, MODBUS-RTU, CC-Link, or CANopen). However, we also include communication standards that are common for industrial networks like OPC UA and MQTT, even though they are not referred to as *classical fieldbusses*.

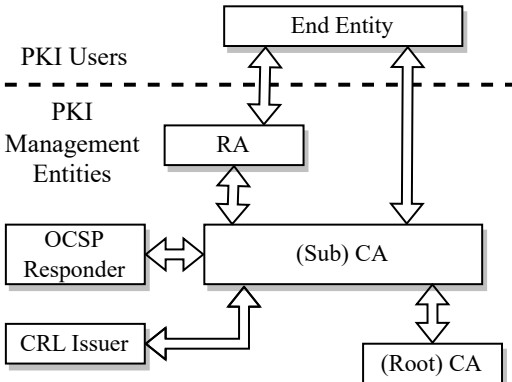

**Figure 1.** The acting entities in a PKI comprise end entities, CAs, and optionally RAs, CRL issuers, and OCSP responder entities. Arrows between entities indicate a management interaction.

*Certification authorities* (CAs) are entities that issue certificates and certificate revocation information. A CA may actually be a "third party" from the EE's point of view. However, quite often, a CA actually belongs to the same organization as the EE. CAs are referred to as "issuers", and their identity is expressed in the *issuer* field of the certificate. We use the term "root" CA to indicate that a CA is the subject and issuer of a certificate at the same time and therefore has a self-signed certificate. A "subordinate" (also referred to as sub

or intermediate) CA is one that is not a root CA for the end entity in question. A sub CA can be certified by a CA to issue certificates to end entities or further sub CAs. In general, CAs are also responsible for indicating the revocation status of the certificates they issue. Therefore, they can use, for example, certificate revocation lists or the Online Certificate Status Protocol (OCSP) [15], which will be described in more detail in Section 4.3. However, a CA may delegate the responsibility to issue revocation information to a different entity. Next to end entities and certification authorities, some environments optionally call for the existence of a registration authority (RA) that is separate from a CA. A registration authority can be an entity to which a CA delegates certain management functions. The functions that are potentially carried out by RAs vary from case to case and may include the authentication of end entities prior to certificate issuance by a CA, the archival of key pairs, or revocation reporting. When an RA is not present, the CA is assumed to be able to carry out the RA's functions so that, from an EE point of view, the management endpoint is the same.

### 3. Survey Methodology

A methodological-formal approach is followed in order to reduce bias due to a selective literature choice. In the following, the phases of the selection process are described to increase transparency and reproducibility (see Figure 2).

Firstly, research questions that shall be answered throughout the review of the literature are defined. For the work at hand, these are:

- **RQ1** *What relevant literature can be found in the area of life-cycle-oriented certificate management in industrial networking environments?*
- **RQ2** *What are the contributions and key proposals of the reviewed work?*
- **RQ3** *What overarching aspects of certificate management can be found to provide a synopsis that compares the reviewed works?*
- **RQ4** *What are existing research gaps?*

With the research questions at hand, four databases were queried in June 2023. The selection of databases was performed by consulting the top used databases for computer science and electrical and electronic engineering ranked by [16]. The selected databases are IEEE Xplore, Science Direct, Scopus, and Springer Link. All selected databases are accessible from the author's institution without further payment and provide the option to use boolean keywords in the submitted query strings. The returned results were sorted by relevance of the according database, and the first 300 most relevant results per database and per query were taken into account. In order to support better reproducibility and avoid manual merging and the removal of duplicates, the search was automated using the APIs provided by the databases. The keywords used are derived from the research questions. The summary of keywords and the corresponding results are shown in Table 1. After the removal of duplicates, a total of 2042 results were imported into the reference manager.

Next, inclusion (see Table 2) and exclusion (see Table 3) criteria were defined to select relevant results. In the first manual title and abstract screening step, the number of relevant results was reduced to 273. In a second full-text screening step, the results were further condensed to 20 results (see Table 4). The number of results per selection step is shown in Figure 3. The remaining results were taken into account for a full-text analysis in order to extract information that would answer the research questions. The extracted information is first categorized and then discussed according to its contribution and main proposals.

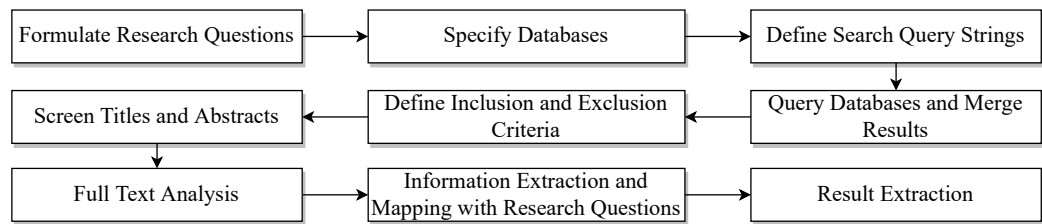

**Figure 2.** A methodological-formal approach is followed in order to reduce bias due to selective literature choice and to increase transparency and reproducibility.

**Table 1.** Search queries and total number of returned results from the selected databases.

| Query | Keywords | IEEE Xplore | Science Direct | Scopus | Springer Link | Total |
|---|---|---|---|---|---|---|
| 1 | "certificate management" AND ("fieldbus" OR "industrial automation" OR "industrial networks" OR "industrial control systems") | 54 | 161 | 76 | 38 | 329 |
| 2 | "credential management" AND ("fieldbus" OR "industrial automation" OR "industrial networks" OR "industrial control systems") | 29 | 7 | 23 | 26 | 85 |
| 3 | ("public key infrastructure" OR "PKI") AND ("fieldbus" OR "industrial automation" OR "industrial networks" OR "industrial control systems") | 143 | 300 | 93 | 300 | 836 |
| 4 | "security life cycle" AND ("fieldbus" OR "industrial automation" OR "industrial networks" OR "industrial control systems") | 66 | 300 | 93 | 26 | 485 |
| 5 | "certificate revocation" AND ("fieldbus" OR "industrial automation" OR "industrial networks" OR "industrial control systems") | 14 | 161 | 4 | 29 | 208 |
| 6 | ("certificate" OR "credential") AND ("deployment" OR "enrollment") AND "industrial" | 14 | 300 | 64 | 300 | 678 |
| | Total | 320 | 1229 | 353 | 719 | 2621 |
| | Total (duplicates removed) | | | | | 2042 |

**Table 2.** Inclusion criteria.

| No. | Inclusion Criteria |
|---|---|
| 1 | The article focuses on the deployment, setup, management, or maintenance of public key infrastructures in industrial environments. |
| 2 | The article focuses on the management of certificates, keys, or trust anchors in industrial environments, especially on IA components. |
| 3 | The article focuses on one or more management procedures of certificates, i.e., the initial deployment, renewal, revocation, or removal. |
| 4 | The article focuses on the life cycle of industrial automation components. |

**Table 3.** Exclusion criteria.

| No. | Exclusion Criteria |
|---|---|
| 1 | The article is not published in English or German. |
| 2 | The article has no focus on the management of certificates, keys, or trust anchors. |
| 3 | The article describes only the usage and not the management of certificates, keys, or trust anchors. |
| 4 | The article describes certificate management avoidance strategies. |

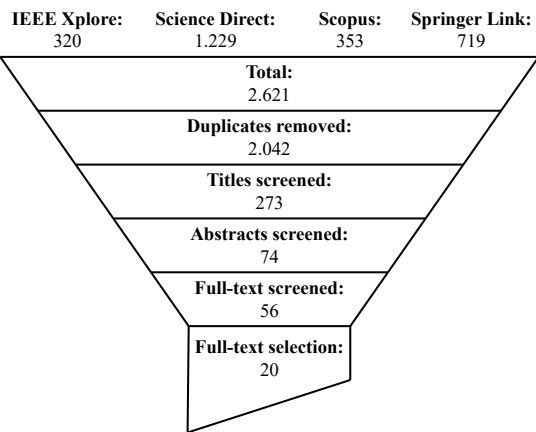

**Figure 3.** Throughout the selection process of papers, the total number of 2042 papers imported into the reference manager was condensed to 20 that were relevant to the topic.

## 4. Presentation of Certificate Management Approaches

This section presents and summarizes the main contributions of the selected approaches that deal with the topic of certificate management in industrial networking environments. We present contributions [11,12,14,15,17–60] from different origin domains: research papers (Section 4.1), industrial communication standards (Section 4.2), and IT domain standards (Section 4.3).

### 4.1. Approaches Presented in Research Papers

The selected research papers address a wide variety of aspects of the comprehensive theme of certificate management in industrial communication environments. This section strives to cluster and summarize the selections with respect to their main contributions. A summary of this subsection is given in Table 4.

Runde et al. [34], Hausmann et al. [30], Fischer et al. [28], Danilchenko et al. [45], Höglund et al. [46], Astorga et al. [11], and Kohnhäuser et al. [51] propose mechanisms for the secure onboarding and initial equipment of devices with certificates issued by the domain of the owner or operator. These mechanisms are secured by using credentials installed on an IA component at manufacturing time. Runde et al. [34] contribute a comprehensive protocol proposal for PROFINET, and Kohnhäuser et al. [51] refer their elaborations to OPC UA.

Kulik et al. [49] and Park et al. [48] both present protocols for the equipment of device credentials without the usage of credentials installed at manufacturing time. Park et al. [48] present comprehensive security extensions for MQTT-SN. Kulik et al. [49] present a formal verification of their protocol proposal.

Fernbach et al. [29], Falk et al. [33], Karthikeyan et al. [41], Krishnan et al. [53], and Yunakovsky et al. [50] present the integration of public key infrastructures in OPC UA environments without a specific focus on the initial secure onboarding. Yunakovsky et al. [50] specifically focus on PKIs in the post-quantum era. Falk et al. [33] propose as an additional element to certificates the usage of certificate whitelists as managed objects.

Garba et al. [58] and Kim et al. [32] both present lightweight mechanisms with the goal of enabling PKIs for resource-deprived IA components. Garba et al. [58] present a lightweight certificate profile, whereas Kim et al. [32] contribute an efficient and scalable protocol proposal for the deployment and revocation of certificates. In a similar manner, Duan et al. [42], Boudagdigue et al. [47,59], and Höglund et al. [57] focus on the revocation of certificates given the circumstances of resource-deprived IA components. In particular, Höglund et al. [57] tailor the mechanism for low power consumption of the IA components.

**Table 4.** Overview of investigated research papers. All of the investigated academic works make conceptual contributions. Additional contributions like a concrete protocol proposal (PP), implementation (IM), formal analysis (FA), experimental measurement (EM), or a mapping to a concrete industrial communication standard (ICS) are marked accordingly.

| Ref. | PP | IM | FA | EM | ICS | Main Contributions |
|---|---|---|---|---|---|---|
| [34] | ✓ | ✓ | - | - | PROFINET | Two-layered certificate architecture, comprehensive security extensions for PROFINET including an extension of the IKEv2 protocol to support the deployment of certificates and trust anchors |
| [30] | - | - | - | - | - | Conceptual proposal of a two-layered certificate architecture issued by (1) the manufacturer of the component and (2) the operator of the component to support the establishment of secure communication and protective measures against product piracy |
| [28] | - | - | - | - | - | Comparison of approaches for secure initial credential bootstrapping using secure device identifiers, requirements for secure device identifiers for authentication |
| [45] | - | ✓ | - | ✓ | - | Conceptual mechanism for an initial security configuration and CA certificate imprinting for IIoT devices in order to minimise manual configuration |
| [29] | - | - | - | - | OPC UA | Comparison of trust models for OPC UA applications based on certificates, certificate life cycle description in PKIs, comparison of PKI frameworks for OPC UA |
| [41] | - | ✓ | - | - | OPC UA | Integration of PKI and access control mechanisms in OPC UA applications, demonstration application with an example PKI architecture |
| [51] | - | - | - | - | OPC UA | Overview, analysis, and comparison of secure OPC UA device provisioning solutions, including initial credential bootstrapping |
| [53] | - | ✓ | - | - | OPC UA | Reference architecture comprising a *Provisioning* and *Device Management Server* with services to bootstrap, provision, renew, and revoke certificates |
| [11] | ✓ | ✓ | - | ✓ | - | Architecture and protocol proposal for an IIoT identity management system based on CoAP and SCEP |
| [58] | ✓ | ✓ | ✓ | ✓ | - | Lightweight self-signed certificate data object proposal, the binding of the identity to the self-signed certificate is validated through the Ethereum blockchain |
| [48] | ✓ | ✓ | - | ✓ | MQTT-SN | Comprehensive security extensions for MQTT-SN including an architecture and protocol proposal to manage certificates on publisher, subscribers, and brokers |
| [46] | ✓ | ✓ | - | ✓ | - | Automated lightweight certificate enrollment protocol for resource-constrained devices, design of a lightweight profile for X.509 digital certificates with CBOR encoding |
| [49] | ✓ | - | ✓ | - | - | Formally verified protocol for system operators and users to manage credentials in ICS |
| [33] | - | - | - | - | - | Usage of explicit certificate whitelisting for the management of device certificates on field-level automation devices, management of certificate whitelists |
| [32] | ✓ | - | - | - | - | Energy-efficient PKI integration scheme for wireless sensor nodes using elliptic curve cryptography |
| [50] | - | - | - | - | - | Security recommendations for PKIs for production environments in the post-quantum era |
| [57] | ✓ | ✓ | - | ✓ | - | Lightweight OCSP alternative using standardized protocols (CoAP and CBOR) and a protocol for compressed CRL distribution |
| [42] | - | ✓ | - | ✓ | - | Parametrizable certificate revocation mechanisms for IIoT devices to find a balance between the occupation of RAM and network bandwidth |
| [47] | ✓ | ✓ | - | ✓ | - | Cluster-based certificate revocation mechanism using game theory mechanisms to renew certificates of nodes that behave well |
| [59] | - | ✓ | - | ✓ | - | IIoT network organization using a clustering architecture, with cluster heads hosting an agent that renews and revokes certificates of cluster members based on signaling game theory |

### 4.2. Approaches Presented in Industrial Communication Standards

The following section presents and summarizes certificate management approaches originating from industrial communication standards. Figure 4 summarizes the findings.

First, in order to have a well-defined selection of industrial communication standards, the market shares for newly connected nodes in industrial networks in 2022 are considered [61]. The total market shares can be split into industrial Ethernet (66%), non-Ethernet-based fieldbus systems (27%), and wireless systems (7%). Among industrial Ethernet, *EtherNet/IP* (25%), *PROFINET* (25%), *EtherCAT* (17%), *Modbus-TCP* (9%), *POWERLINK* (5%), and *CC-Link IE Field* (3%) make up a share of nearly 85%. The non-Ethernet-based fieldbus systems are composed of *PROFIBUS DP* (26%), *Modbus-RTU* (19%), *CC-Link* (15%), *DeviceNet* (15%), and *CANopen* (7%). Due to their low total market shares, wireless industrial networks are not further taken into account. Besides the mentioned Ethernet and non-Ethernet-based industrial networks, two other (draft) standards were taken into consideration: *OPC UA* [62] and the time-sensitive networking (TSN) profile defined by the current draft version 2.0 of *IEC/IEEE 60802* [63]. *OPC UA* serves as a protocol for monitoring and controlling industrial processes [64]. Therefore, it is included in the selection for further analysis. The TSN profile for industrial automation is a joint project of IEC and IEEE and is currently being released in a draft version.

Second, with the selected industrial communication standards at hand, they are categorized into standards that (a) don't specify the usage of certificates (including PROFIBUS DP [65], MODBUS-RTU [66], CC-Link [67], DeviceNet [68], CANopen [69], CC-Link IE Field [70], POWERLINK [71], and EtherCAT [72]); (b) specify the usage of certificates but don't specify the management of certificates (including MODBUS-TCP [73]); and (c) also specify the management of certificates (including EtherNet/IP with Common Industrial Protocol (CIP) Security [74], PROFINET [52], OPC UA [62], and IEC/IEEE 60802 [60]).

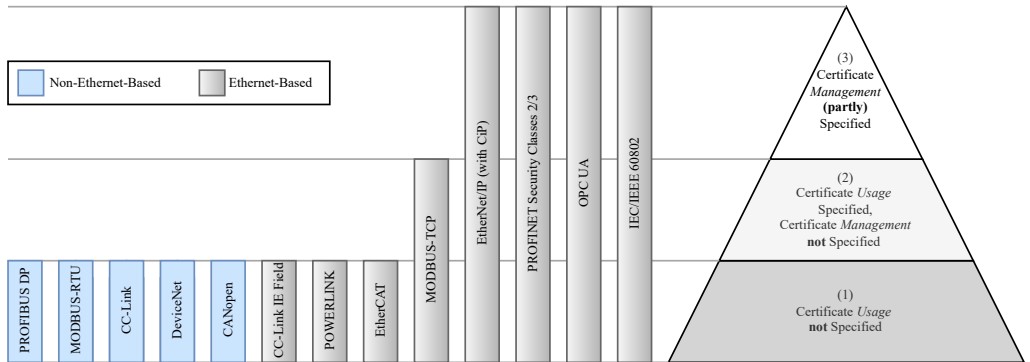

**Figure 4.** Classification of Ethernet-based and non-Ethernet-based industrial communication standards with regard to their specification of usage and management of certificates.

Third, the industrial communication standards that are categorized as belonging to (c) are considered in more detail. The remainder of this section strives to summarize the certificate management approaches presented in *EtherNet/IP with CIP Security*, *PROFINET*, *OPC UA*, and *IEC/IEEE 60802* TSN profile for industrial automation.

The first **EtherNet/IP** specification was released in 2001 [18]. In 2007, EtherNet/IP was first published as an IEC standard in IEC 61158 and IEC 61784. However, the security extensions, incorporated in EtherNet/IP with CIP Security, were first published in 2015 [37]. The authors of the work at hand do not have access to the CIP Security specification. The gathered information relies on publicly available documents from the ODVA and vendors of CIP-Security-enabled devices [37]. Certificate management in CIP Security comprises a *Certificate Management Object* (CMO) that manages X.509 certificates maintained by the device and manageable by a commissioning application [75]. Moreover, CIP Security also specifies the usage of the EST protocol [27] in order to let CIP-Security-enabled devices pull certificates from a CA.

Similar to EtherNet/IP, in its initial version, **PROFINET** did not include cryptographic protection mechanisms in the protocol specification. Since version 2.4 MU2, which was released in 2021, the PROFINET specification defines the supply and management of security configuration parameters, including site-specific X.509 certificates, private keys, and trust anchors to PROFINET system components. The approach comprises a Security Infrastructure Handler (SIH) implementing the functionality to push certificates, private keys, and trust anchors to PROFINET system components using the Site Credential Management class and a PROFINET-native protocol to talk to PROFINET system components [52]. In addition to the push-supply of private keys, a PROFINET system component can also be capable of generating a private key itself and to form an according Certificate Signing Request (CSR) that is sent to an SIH. The term push-supply refers to the interaction model between the SIH that actively triggers actions by sending requests to PROFINET system components that react and respond accordingly. The authors of the work at hand have access to the above-mentioned PROFINET specification and participate in the security working group of PROFINET.

Certificate management in **OPC UA** systems is addressed in several parts of the specification. In the following, we refer to the latest releases of the OPC UA specification [54–56] that are available online. In OPC UA Part 2 [54], the security model is described, which also comprises a general overview of certificate management. Moreover, in Part 12 [55], the functionality of a *GlobalDiscoveryServer* (GDS) that manages certificates is specified. Both parts have been updated regularly since their first publication. In the recently published Part 21 [56] from 2022, the onboarding of OPC UA devices is specified, which also comprises certificate management-related aspects as explained in the further sections.

The following considerations about the **IEC/IEEE 60802** TSN profile for industrial automation refer to the online available draft version D2.0 [60]. The profile specification presents the *security setup* of industrial automation (IA) components occurring in different *life cycle phases* of the IA component. The draft standard describes the utilization of NETCONF/YANG over TLS [76] to enable the management of certificates, keys, and trust anchors [60]. TSN domain management entities (TDME) are responsible for the TSN network, acting as NETCONF clients, and are able to push certificates, keys, and trust anchors onto IA components, acting as NETCONF servers.

*4.3. Approaches Presented in Standards Predominantly Originating from the IT Domain*

The following section presents and summarizes certificate management approaches originating from the IT domain. The rationale behind this presentation is that some proposals from Sections 4.1 and 4.2 integrate IT domain protocols into their proposals in order to set up a communication and security relation with CAs hosted by an owner or operator of an industrial networking environment. This enables the use of already-existing infrastructure for certificate management. For this purpose, we want to provide an overview of IETF RFCs dealing with the topic of certificate management. Table 5 summarizes the findings.

The *Certificate Management Protocol* (CMP), specified in IETF RFC 4210 [14], and the *Certificate Management over CMS* (CMC) protocol, specified in IETF RFC 5272 [19], both present management protocols that enable the deployment of keys, trust anchors, and private keys, as well as the renewal and revocation of certificates. The *Trust Anchor Management Protocol* (TAMP), specified in IETF RFC 5934 [24], describes a transport-independent protocol limited to the management of trust anchors, and their renewal, removal, and revocation.

Even though it does not specify a certificate management protocol, the *Internet X.509 Public Key Infrastructure Certificate and CRL Profile*, specified in RFC 5280 [12], profiles the X.509 v3 certificate and X.509 v2 certificate revocation list (CRL) for their use in the Internet. This profile serves as the underlying basis for standards like the IEEE 802.1 AR [26], where initial device identifiers (IDevIDs) and locally significant identifiers (LDevIDs) are specified, and for the certificate profile of the industrial communication standards PROFINET, IEC/IEEE 60802, CIP Security, and OPC UA, and various approaches from research papers. Specifically focusing on the enrollment of devices, the *Enrollment over Secure Transport* (EST)

protocol, specified in IETF RFC 7030 [27], the *Simple Certificate Enrollment Protocol* (SCEP), specified in IETF RFC 8894 [17], and the *Enrollment with Application Layer Security* (EALS) draft [40] leverage existing technologies like CMS, HTTPS, CoAP, and CBOR to equip clients with trust anchors and certificates.

With the goal of radically simplifying the deployment of HTTPS in the web context, the *Automatic Certificate Management Environment* (ACME), specified in IETF RFC 8555 [39], describes a protocol by which an applicant can obtain a signed certificate from a CA. The *Bootstrapping Remote Secure Key Infrastructure* (BRSKI) protocol, specified in IETF RFC 8995 [35], provides a solution for the automated bootstrapping of a remote secure key infrastructure of new (unconfigured) devices using manufacturer-installed X.509 certificates.

Lastly, the *Online Certificate Status Protocol* (OCSP), specified in IETF RFC 6960 [15], and the *TLS Certificate Status Request Extension* (also referred to as OCSP Stapling), specified in IETF RFC 6066 [77], both focus on the revocation of certificates.

**Table 5.** Summary of certificate management approaches originating from the IT domain.

| Ref. | Name | Main Contribution |
|---|---|---|
| [14] | CMP: Certificate Management Protocol | Describes protocol messages for all relevant aspects of certificate creation and management |
| [12] | Internet X.509 Public Key Infrastructure Certificate and CRL Profile | Profiles X.509 Certificates and CRLs for use in the Internet |
| [19] | CMC: Certificate Management over CMS | Base syntax for a certificate management protocol enabling clients to acquire EE and CA certificates from a CMC server |
| [24] | TAMP: Trust Anchor Management Protocol | Describes protocol messages to manage trust anchors on devices using CMS |
| [15] | OCSP: Online Certificate Status Protocol | Specifies the message exchanges between an OCSP client and responder entity to retrieve revocation status information about a certificate in a more timely way compared to CRL distributions |
| [31,77] | OCSP Stapling, TLS Multiple Certificate Status Request Extension | Allows one to convey status information from the server's copy of certificate status information directly in the TLS handshake messages, saving round-trips |
| [27] | EST: Enrollment over Secure Transport | Profiles certificate enrollment using CMC over secure transport to acquire client and CA certificates |
| [39] | ACME: Automated Certificate Management Environment | Automates the process to verify that an applicant for a certificate legitimately represents the domain name and provides facilities for certificate management functions |
| [17] | SCEP: Simple Certificate Enrollment Protocol | Enables PKI client to request or renew certificates and CRLs using CMS and PKCS #10 over HTTP |
| [35] | BRSKI: Bootstrapping Remote Secure Key Infrastructure | Enables automated bootstrapping of a new (unconfigured) device in order to securely deploy the new key infrastructure to the device |
| [40] | EALS: Enrollment with Application Layer Security | IETF draft of enrollment procedures leveraging CoAP, CBOR, or COSE with application-layer security specifically designed for IoT devices |

An overview and timeline of the publications are presented in Figure 5. The timeline reflects the different origin domains in the three columns: research papers (Section 4.1) in the blue column, industrial communication standards (Section 4.2) in the red column, and IT domain standards (Section 4.3) in the green column. If there are previous releases of industrial communication standards, they are marked in dotted rectangles. For the sake of simplicity, not all release versions are shown, but only the first publication, the latest one that we are referring to, and, if applicable, intermediate released version that are referred to by other contributions. Since some research papers refer to IETF RFC drafts, we also listed the first appearance of the draft standards in dotted rectangles. An arrow from contribution A to contribution B indicates that A uses techniques from, cites, or references B. To improve the clarity of the graphic, arrows from predecessor to successor versions of industrial standards and from IETF drafts to RFCs have been omitted. The only dotted arrow drawn in the timeline reflects that an earlier published research paper references a contribution that was still unpublished.



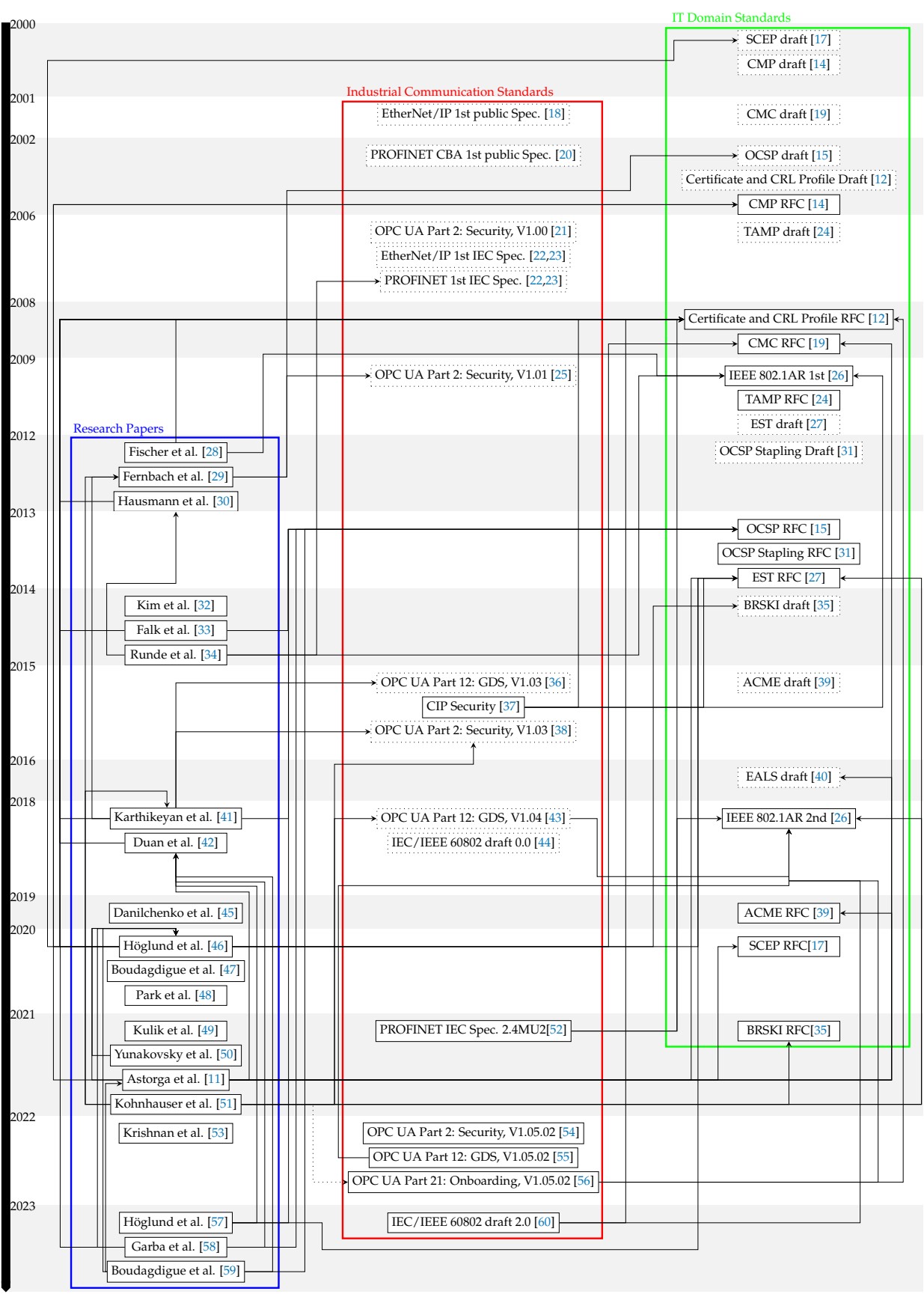

**Figure 5.** Significant contributions to certificate management in industrial networking environments [11,12,14,15,17–60]. An arrow from contribution A to B indicates that A uses techniques from, cites, or references B.

## 5. Synopsis of Certificate Management Approaches

This section provides a synopsis of the approaches presented in Section 4. Therefore, we introduce overarching aspects to present a system structure and comparison of the individual approaches. These overarching aspects are *entity reference architectures, considered functions, involved stakeholders*, and *life cycle stages*. Our understanding of an *entity reference architecture* is the description of involved entities, their purpose, and how they interplay with each other. As *functions*, we understand the set of certificate management functionality considered by a proposal (e.g., the issuance of a certificate or revocation information). Some proposals discuss the involvement of different stakeholders, for example manufacturers, system integrators, or operators of IA components. As *life cycle stages*, we understand stages in the life cycle of the IA components that deserve certificate-management-related treatments. Table 6 summarizes the comparison and system structuring.

### 5.1. Entity Reference Architectures

The studied works that we presented in Section 4 incorporate three different types of entity reference architectures that differ in the involved entities, their purpose, and their interplay. In Figure 6, the three different types are visualized. In the following, we present these types and assign the studied works to the different architectures. The classification of the approaches is given in Table 6 in the first column.

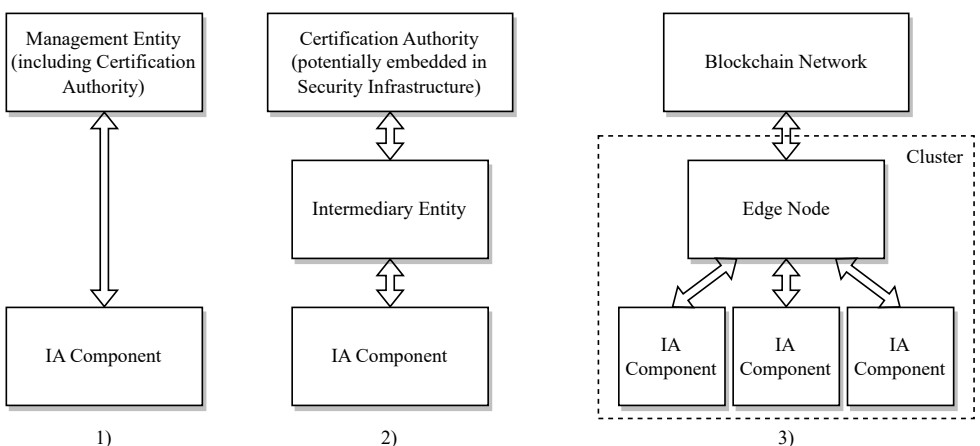

**Figure 6.** Three main reference architectures of the involved entities and their communication relations for certificate management.

The first type considers the most simple entity reference architecture, only comprising the to-be-managed IA component and a management entity. The underlying assumption of this architecture is that an IA component and a management entity provide common protocol mechanisms to proceed with the certificate management processes. Moreover, since no further entity is involved in the process, a management entity is capable of issuing certificates and trust anchors, which therefore inheres the purpose that is classically ascribed to CAs.

Fischer et al.'s [28], Runde et al.'s [34], Park et al.'s [48], Krishnan et al.'s [53], and Hausmann et al.'s [30] proposals follow this architecture, where this management entity is located on an engineering tool and, for example, deployed on a laptop or tablet and operated by a human user for PROFINET components. This engineering tool features the option to generate a root CA certificate, used as a trust anchor by the IA components, and to issue end entity certificates.

Fernbach et al.'s [29], Karthikeyan et al.'s [41], and Höglund et al.'s [46] proposals slightly differ from the previously mentioned proposals by providing a more fine granular resolution of the management entity, such that it is not composed of a singular entity but consists of a registration authority (RA) that verifies certificate management requests, a CA

that issues certificates, and a validation authority (VA) that validates certificates. This more fine-grained picture of the management entity matches the description of PKI entities as described in the IETF RFC 4210 [14] and 5280 [12] (see Section 2).

When analyzing the entity architectures from IT-domain-based proposals, we observe this direct interaction between the to-be-equipped entity and the management entity in CMP [14], CMC [19], TAMP [24], EST [27], ACME [39], and SCEP [17]. In addition, the revocation IETF RFCs specifying OCSP [15] and OCSP stapling [77] describe a direct interaction between the entity that wants to receive revocation information and the responder that distributes the information.

The second type of reference architecture incorporates three entities: an IA component, an intermediary entity, and a CA optionally embedded in further security infrastructure, potentially with an RA or a VA. The additional intermediary entity acts as the direct interaction partner with the IA component and is capable of executing required services for certificate management (e.g., the generation of an asymmetric key pair or imprinting a signed certificate). The intermediary entity supports the protocol (i.e., syntax and semantics) of the IA component. Since IA components likely only support industrial communication protocols and at the same time CAs likely only support IT-domain-based certificate management protocols (as presented in Section 4.3), it is the intermediary entity's task to supply certificate management commands into industrial protocol native forms. Thus, it bridges the gap between IT-domain-based and industrial communication protocols. Additionally, it may act as a delegated entity that orchestrates certificate management processes.

As proposed by Astorga et al. [11], such an intermediary entity may be located at the boundary between IA components and the corporate network, and by delegating certificate management tasks to this entity, the complexity associated with certificate management processes is moved from the resource-deprived IA components to a more resource-rich edge device. In addition to the intermediary entity's communication relations to IA components, Astorga et al.'s [11] intermediary entity also features a second type of communication relation to already-existing corporate security infrastructure, comprising the SCEP protocol for the automatic (re)enrolment of certificates. Falk et al.'s [33], Duan et al.'s [42], Fischer et al.'s [28], and Danilchenko et al.'s [45] proposals all incorporate different entities for the generation and distribution of security configuration artifacts.

In the current specification of security extensions for PROFINET, a *Security Infrastructure Handler* (SIH) can have the role of the intermediary entity, executing certificate management processes. This enables an SIH to equip PROFINET system components with certificates in a push-supply manner. However, the specification does not strictly specify how the SIH gets a certificate in order to supply it to PROFINET system components. Conceptually, both entity architectures, the first and the second, are feasible. The SIH may either feature a co-located (sub) CA or have another communication relation to already-existing CA infrastructure that enables integration into legacy infrastructure using, for example, one of the protocols mentioned in Section 4.3. This also applies for IEC/IEEE 60802, where the TDME uses NETCONF/YANG to interact with IA components to proceed with credential management tasks. The management enables a push supply of certificates, keys, and trust anchors. Likewise, OPC UA provides OPC-UA-specific protocol mechanisms for certificate management functionality using the *CertificateManager* in conjunction with the *GlobalDiscoveryServer* (GDS). The GDS may act in a push-supply mode or OPC UA applications may pull certificates or trust lists from the GDS. Comparably, CIP Security also provides CIP-specific protocol mechanisms to manage certificates by using Certificate Management Objects (CMOs). In commissioning applications, for example, a configuration tool may request the CMO of a device to create a CSR and can obtain a certificate, either local to the application or reachable over network means, that is then further pushed to the CIP Security device. In addition, CIP Security also supports a pull model that allows devices to pull certificates directly from CAs using EST. In IT-domain-based standards, we observe the mention of intermediaries in the BRSKI RFC [35] and EALS draft [40], both introducing proxy components for the initial bootstrapping process.

The third type of entity reference architecture incorporates cluster-based architectures and the use of blockchains. Boudagidgue et al. [47,59] propose dynamically forming clusters of IIoT devices consisting of a set of member nodes and one cluster head. Cluster heads act as CAs that also participate in a blockchain network that shares information about the member nodes' behavior. Based on the behavior of the member nodes, they are provided with certificates or not. Garba et al. [58] similarly propose the usage of blockchain networks. IIoT devices shall issue self-signed certificates and let local registration authorities verify and validate the identity binding through the Ethereum blockchain that is used as a global notary of certificates.

To conclude, we observe three main types of entity reference architectures that the reviewed works are based on. Certificate management approaches presented in industrial communication standards are based on the first or second architecture. None of them propose the use of cluster- or blockchain-based approaches. Moreover, all reviewed industrial communication standards present their protocol-specific mechanisms to proceed with certificate management processes. Only CIP Security also specifies the usage of EST for CIP-Security-enabled devices. These commonalities give rise to the development of a unified generic model that describes certificate management for IA components from existing industrial standards in a protocol-independent manner.

### 5.2. Certificate Management Functions

In this subsection, we discuss certificate management functions that are addressed and supported by the approaches from Section 4. Our understanding of a certificate management function refers to a functionality that leads to changes regarding the cryptographic artifacts on IA components that were introduced within the presentation of background knowledge in Section 2. We elaborate in Section 5.4 when these functions are executed within the life cycle of IA components. As discussed in the presentation of background knowledge in Section 2, IA components require certificates, private keys, and trust anchors to achieve the security goal of mutual authentication. Therefore, functions are required to initially deploy these cryptographic elements on IA components. However, as the following subsections show, not only the initial deployment of certificates is necessary, but also their renewal, removal, and revocation. The synopsis of supported, addressed, or mentioned functions is given in Table 6 from the second to the sixth column.

#### 5.2.1. Equipment with Trust Anchors

First, we consider the deployment of an owner- or operator-specific trust anchor that is required to authenticate other devices, (logical) entities, or users of the domain of the owner or operator, as described in Section 2.

Runde et al. [34], Karthikeyan et al. [41], and Astorga et al. [11] introduce functions to set trust anchors for their implementations. All approaches use certificates as trust anchors, while only Astorga et al. [11] propose the usage of a raw public key. In addition, Falk et al. [33] highlight the importance of explicitly configuring trusted (CA) certificates as trust anchors.

In industrial communication standards, each of the reviewed approaches supports an explicit service to set trust anchors in IA components. IEC/IEEE 60802 and the latest PROFINET specification describe the *imprinting* of a trust anchor in the form of a certificate as a part of the security setup after booting from the factory default state. The term imprinting refers to equipping IA components with certificates, private keys, and trust anchors. Similarly, OPC UA systems use a GDS with certificate management functionality to maintain a list of trusted (CA) certificates (TrustList) on the IA component, and CIP Security supports the setting of trusted CA certificates using the CMO [37].

**Table 6.** Synopsis of entity reference architectures (ERA, numbers according to Figure 6), addressed functions, involvement of stakeholders (SH, including manufacturer (M), component builder (CB), distributor (D), integrator (I), and owner/operator (OO)), and life cycle stages (LCS, including manufacturing (M), onboarding (ON), operational (O), and decommissioning (D)) from approaches presented in Section 4. ✓, indicates that a function is supported, addressed, or mentioned in the work; -, indicates that this aspect is not explicitly mentioned or specified.

| Ref. | Name | ERA | TA Equip. | PK Equip. | Cert. Equip. | Ren. | Rem. | Rev. | SH | LCS |
|---|---|---|---|---|---|---|---|---|---|---|
| [34] | Runde et al. | 1 | ✓ | int. | ✓ | - | - | - | M,OO | M,ON,O |
| [30] | Hausmann et al. | 1 | - | - | ✓ | - | - | ✓ | M,OO | M,ON,O |
| [28] | Fischer et al. | 1,2 | ✓ | both | ✓ | - | - | ✓ | M,OO | M,ON |
| [45] | Danilchenko et al. | 2 | ✓ | ext. | ✓ | ✓ | - | ✓ | M,OO | M,ON |
| [29] | Fernbach et al. | 1 | - | - | ✓ | ✓ | - | ✓ | OO | O |
| [41] | Karthikeyan et al. | 1 | ✓ | int. | ✓ | - | - | ✓ | OO | O |
| [51] | Kohnhäuser et al. | - | - | - | ✓ | - | - | - | M,OO | M,ON |
| [53] | Krishnan et al. | 1 | - | both | ✓ | ✓ | - | ✓ | OO | ON,O |
| [11] | Astorga et al. | 2 | ✓ | both | ✓ | ✓ | - | ✓ | M,OO | M,ON,O |
| [58] | Garba et al. | 3 | - | int. | ✓ | ✓ | - | ✓ | OO | ON,O |
| [48] | Park et al. | 1 | - | ext. | ✓ | ✓ | - | ✓ | OO | ON,O |
| [46] | Höglund et al. | 1 | - | both | ✓ | ✓ | ✓ | ✓ | M,OO | M,ON,O,D |
| [49] | Kulik et al. | 1 | - | - | - | - | - | - | OO | O |
| [33] | Falk et al. | 2 | ✓ | - | ✓ | - | - | ✓ | OO | O |
| [32] | Kim et al. | 1 | - | int. | ✓ | ✓ | - | ✓ | OO | O |
| [50] | Yunakovsky et al. | - | - | - | ✓ | - | - | ✓ | OO | O |
| [57] | Höglund et al. | 1 | - | - | - | - | - | ✓ | OO | O |
| [42] | Duan et al. | 2 | - | - | - | - | - | ✓ | OO | O |
| [47] | Boudagdigue et al. | 3 | - | - | - | ✓ | - | ✓ | OO | O |
| [59] | Boudagdigue et al. | 3 | - | - | - | ✓ | - | ✓ | OO | O |
| [54–56] | OPC UA | 1,2 | ✓ | both | ✓ | ✓ | ✓ | ✓ | M,CB,D, I,OO | M,ON,O,D |
| [37] | CIP Security | 1,2 | ✓ | both | ✓ | ✓ | ✓ | - | M,OO | M,ON,O,D |
| [52] | PROFINET | 1,2 | ✓ | both | ✓ | ✓ | ✓ | - | M,OO | M,ON,O,D |
| [60] | IEC/IEEE 60802 | 1,2 | ✓ | both | ✓ | ✓ | ✓ | - | M,OO | M,ON,O,D |
| [14] | CMP | 1 | ✓ | both | ✓ | ✓ | - | ✓ | - | ON,O |
| [12] | RFC 5280 | 1 | - | - | ✓ | - | - | ✓ | - | - |
| [19] | CMC | 1 | ✓ | both | ✓ | ✓ | - | ✓ | - | ON,O |
| [24] | TAMP | 1 | ✓ | - | - | ✓ | ✓ | ✓ | - | O |
| [15] | OCSP | 1 | - | - | - | - | - | ✓ | - | O |
| [31,77] | OCSP Stapling | 1 | - | - | - | - | - | ✓ | - | O |
| [27] | EST | 1 | ✓ | - | ✓ | ✓ | - | - | - | ON,O |
| [39] | ACME | 1 | - | - | ✓ | ✓ | - | ✓ | - | ON,O |
| [17] | SCEP | 1 | ✓ | - | ✓ | ✓ | - | ✓ | - | ON,O |
| [35] | BRSKI | 2 | ✓ | - | ✓ | ✓ | ✓ | - | M,OO | M,ON,D |
| [40] | EALS | 2 | - | - | ✓ | - | - | - | - | ON |

Regarding IT-domain-based standards, some IETF RFCs explicitly specify the equipment with trust anchors. This includes the TAMP [24]. Although not mentioned explicitly as a trust anchor, in CMP [14], during the initialization of the end entity that is subject to management operations, it needs to securely acquire a copy of the relevant root CA public keys by out-of-band means. If the CA and the end entity use "shared secret information", then an initialization response message may contain certificates that can be directly trusted. The CMC [19] has a dedicated "Publish Trust Anchor" control to distribute a set of trust anchors to end entities in the form of certificates. Similarly, the EST [27] protocol and SCEP [17] support the distribution of CA certificates to be used as trust anchors. Moreover, BRSKI [35] provides an automated mechanism to distribute CA certificates in an authorized manner.

To conclude, each industrial communication standard describes the equipment with CA certificates that can be used as trust anchors to validate certificates originating from the domain of the owner or operator of the IA component. In addition, standards originating from the IT domain either describe functions to supply CA certificates or require that IA components are already equipped with them and do not specify the means to do so. Very few research papers deal with the topic of trust anchor equipment.

### 5.2.2. Equipment with Private Keys

Next, we consider the equipment of IA components with private keys. The issuance of a certificate requires the existence of a public key that is going to be cryptographically bound to identity information by the certificate. Therefore, prior to the issuance of a certificate, the generation of an asymmetric key pair is required. Here, we observe two possibilities as to which entity performs the key pair generation: an IA component may either internally generate and store an asymmetric key pair or an external entity generates a key pair and conveys it to the IA component for storage and usage. This can be useful if the capabilities of an IA component, due to some constraints, do not allow the internal generation of a key pair. In the event that the IA component internally generates the asymmetric key pair, it subsequently generates a certificate signing request (CSR) that is sent to the certificate management interaction partner conveying the public key portion to obtain a signed certificate.

The equipment with private keys is considered by some research papers that address both possibilities [11,28,46,53], only the device internal key generation [32,34,41,58], or only external key generation [45,48].

Moreover, all of the addressed industrial communication standards from Section 4.2 support both possibilities where the asymmetric key pair is generated. Regarding IT-domain-based standards, the CMP [14] requires that the generation of key pairs may occur elsewhere and not only in end entities, and CMC [19] requires that all key generation can occur on the client, but also states that RAs may perform additional services such as key generation.

### 5.2.3. Equipment with Certificates

The approaches presented in Section 4 propose different possibilities for how IA components are equipped with certificates that are used to authenticate to entities from the domain of the owner or operator. In the most general way, certificates are issued by certification authorities upon receiving a CSR. Once a certificate is issued, the signed certificate is sent to the component it was issued for.

We observe this sequence in research papers from Fischer et al. [28], Hausmann et al. [30], Karthikeyan et al. [78], Kim et al. [32], Park et al. [48], and Krishnan et al. [53]. Some approaches detail this function even further, for example, Runde et al. [34] present a dedicated service to convey a certificate that was signed by a CA via an engineering tool that is then stored on the IA component. Astorga et al.'s approach [11] also follows the above-mentioned sequence and defines PKCS#10 as the to-be-used data structure to convey the certificate signing request. Fernbach et al. [29] describe different methods of certificate

distribution: manual transportation on a storage medium (e.g., via USB-stick), publication in a well-known public repository (e.g., LDAP), or in-band distribution using a specific communication protocol (e.g., TLS). Höglund et al.'s approach [46] follows this sequence using the *simple enroll request* from EST. Danilchenko et al.'s [45] proposal differs from the others to the effect that IA components are already at manufacturing time equipped with private keys and signed certificates recognized by the device management entity, namely Security Profile Distribution Service (SPDS). At bootstrapping time, the IA component is only equipped with the site's TLS-inspecting proxy CA certificate. In addition, Garba et al. [58] propose a blockchain-based approach that differs from the above-mentioned sequence because devices themselves generate self-signed certificates and an Ethereum blockchain is used as a global notary of certificates.

All industrial communication standards have in common that they do not specify exactly which entity, tool, or mechanism shall issue certificates. As discussed in Section 5.1, this entity may be located at some configuration tool or may be a CA that is embedded in a security infrastructure. However, all protocols describe protocol-specific mechanisms to equip IA components with certificates.

Moreover, the equipment of devices with certificates is subject to IT-domain-based standards. CMP [14] addresses this with the initial registration and certification, whereby an end entity first makes itself known to a CA or RA, a CA issues a certificate, and, as an end result, the CA returns the certificate to the end entity or posts it to a public repository. In CMC [19], the PKCS#10 certificate signing request is used to request a certificate, which is further conveyed to the EE using CMS. Similarly, EST [27], ACME [39], SCEP [17], BRSKI [35], and the EALS [40] draft provide means to equip end entities with certificates.

### 5.2.4. Renewal

Typically, the validity of certificates is bound by an expiration date. For example, X.509 certificates are only valid within the time interval of their (*notBefore*) and (*notAfter*) fields [12]. Therefore, a function is required to replace a certificate with a newly issued certificate that has a longer validity period. As only a small number of works explicitly address the renewal, we address the renewal of certificates, private keys, and trust anchors in one subsection. The renewal function differs from the initial certificate equipment in the possibility of not repeating the potentially time-consuming manual onboarding procedure, as explained in Section 5.4.

Most of the research papers presented in Section 4.1 do not mention or address the renewal of certificates, private keys, or trust anchors at all [28,30,33,34,41,42,50,51]. Fernbach et al. [29], Danilchenko et al. [45], and Garba et al. [58] mention the necessity of renewing certificates but do not explicitly describe a function or mechanism for how the renewal of certificates proceeds. Krishnan et al. [53], Höglund et al. [46], Park et al. [48], Kim et al. [32], and Astorga et al. [11] explicitly describe a renewal function using a specified protocol. Krishnan et al. [53], Park et al. [48], and Kim et al. [32] propose a self-defined protocol, whereas Hoglund et al. [46] propose the usage of EST's re-enrollment function [27] and Astorga et al. [11] the usage of SCEP [17]. Krishnan et al. [53] propose that the device itself be capable of recognizing the necessity of renewing its certificate, whereas Astorga et al. [11] mention that IIoT devices often do not have an understanding of the actual world-time and therefore let their proposed management entity trigger the renewal of certificates. In Boudagdigue et al.'s first publication from 2020 [47], their core idea is to only renew certificates of entities that show good behavior, which is observed by the blockchain network. Moreover, in Boudagdigue et al.'s second publication from 2023 [59], they propose a validity period that is proportional to the trust score to balance the trade-off between the renewal overhead and a high security level.

In OPC UA [54–56], we distinguish between push and pull models with regard to renewal. For the pull model, the OPC UA client is responsible for ensuring that certificates and trust lists are kept up-to-date. The client can request updates for its own certificates and trust lists. For the push model, the certificate manager can call methods to update

the certificates and trust lists as required. In addition, the certificate can be updated without generating a new private key pair. From our available sources that describe CIP Security [79–81], we observe that renewal is possible either using the push management with the CMO or using the EST-specific service to re-enroll. PROFINET and IEC/IEEE 60802 do not specify a dedicated renewal command and do not address the renewal of certificates in their specifications [52,60]. However, since both specifications realize push-management behavior, with the existing services, a renewal can be realized using the function to equip a component with an end entity certificate. This is also possible with or without the renewal of the trust anchor and with or without the renewal of the private key. Similarly, in IEC/IEEE 60802, there is also no dedicated renewal command. The renewal of certificates is possible by reusing the equipment functions. Moreover, the draft specifies that component manufacturers are not required to provide a feature to update IDevIDs credentials. As Section 5.4 will detail, PROFINET, IEC/IEEE 60802, and OPC UA specify that devices are provisioned with an additional dedicated certificate used for mutual authentication with a management entity. This enables the establishment of a secured communication channel for the security configuration including the seamless renewal of certificates over network means.

Regarding IT-domain-based standards, a dedicated function to renew a certificate is specified by CMP [14], EST [27], SCEP [17], and TAMP for trust anchors [24]. Moreover, BRSKI [35] describes the renewal of voucher artifacts. Additionally, CMP [14], EST [27], and SCEP [17] specify that a new certificate may also be issued for the key pair that was in use for the to-be-renewed certificate. In contrast to that, RFC 5280 [12], CMC [19], and ACME [39] only mention the necessity of renewing certificates and do not specify a dedicated function.

To conclude, we observe that some approaches handle the renewal of certificates by reusing the equipment functions, whereas others introduce dedicated renewal or update functions. Moreover, we observe that different flavors of certificate renewal are possible: with or without private key renewal and with or without trust anchor renewal. Furthermore, there are different options for which entity triggers the update process: the device itself or the management entity. This is also linked to the question of whether the device itself is capable of actively triggering certificate management functions and if it has an understanding of the correct world-time.

### 5.2.5. Removal

IA components typically have a limited operating period. An owner or operator of a component may decide to re-sell or discard IA components at the end of their operating period. However, owners or operators do not want to give away their components with their certificates, keys, and trust anchors. Therefore, a function to remove the artifacts before discarding or reselling them is necessary.

However, this is not addressed by any research paper except Höglund et al.'s publication from 2020 [46]. The authors mention that the overwriting of pre-installed CA certificates would introduce additional complexity because they need to be restored when the component is reset to the factory default settings.

In contrast to this, all industrial communication standards provide mechanisms to remove certificates, keys, and trust anchors from the components. In OPC UA, when using push management, there is a "ResetToServerDefaults" method that deletes, among other elements, all trust lists, certificates, and keys. Thus, if the device supports the Part 21 onboarding model [56], it has to restart the onboarding process. Additionally, in the push model, with the "Remove Certificate" method, it is possible to remove single certificates from the trust list. CIP Security also supports mechanisms to delete the CMO and reset a CIP-Security-enabled device back to its factory default settings [75]. This is handled similarly in PROFINET [60] and IEC/IEEE 60802 [60].

In IT-domain-based standards, only TAMP [24] and BRSKI [35] mention the removal of artifacts. TAMP allows the removal of trust anchors. BRSKI mentions two types of resets

to the factory default state. The first is a harder reset that removes all certificates, keys, and trust anchors, and the second is a weaker reset that does not remove trust anchors.

### 5.2.6. Revocation

Under normal circumstances, issued certificates are used until the end of their validity period. However, there are causes that lead to cases where certificates become invalid prior to their expiration [12]. This includes, for example, the change of an identity attribute, the change of the association between the subject and the CA, or the compromise of the corresponding private key. These circumstances require the revocation of certificates [12], which can be subdivided into two steps: first, for example, the intermediary entity requests the revocation of a certificate by a CA, and second, the subsequent distribution of the revocation information. The presented approaches from Section 4 either address both aspects, only one of them, or none at all.

Regarding the studied research papers, Fernbach et al. [29], Karthikeyan et al. [41], Falk et al. [33], Hausmann et al. [30], and Yunakovsky et al. [50] all propose the usage of CRLs and OCSP. Moreover, Hausmann et al. [30] note that finding a suitable solution for certificate revocation is a design challenge related to PKIs in automation systems. Yunakovsky et al. [50] note that revocation lists shall be maintained and updated on a regular basis and that PKI hierarchies shall be developed that allow one to precisely revoke certificates for a specific set of devices.

On the other hand, Krishnan et al. [53], Park et al. [48], and Kim et al. [32] all propose denying the network access to a device by sending its identity to a central entity that can control either access to the network itself or access to get the required certificate to access the network, which compared to classical mechanisms like CRLs or OCSP avoids certificate revocation rather than proposing a solution. In detail, Krishnan et al. [53] propose a request to the Device Management Server that adds the serial number of the device to a list that enumerates devices that will not be able to connect to the DMS again.

Moreover, Astorga et al. [11], Garba et al. [58], Danilchenko et al. [45], Höglund et al. [46,57], Duan et al. [42], and Boudagidgue et al. [47,59] all address shortcomings of either CRLs or OCSP and therefore propose alternative approaches. These shortcomings are the resource-expensiveness regarding memory and bandwidth consumption and potentially complex management overhead. Astorga et al. [11] state that the use of CRLs and OCSP has been demonstrated to be highly resource expensive. As an alternative, they claim that the use of short-lived certificates eliminates the need for managing and checking revoked certificates and enhances overall network security. However, short-lived certificates require the availability of timing information in IA components. Garba et al. [58] propose the usage of LightCerts that realize revocation by updating the record in the smart contract of the Ethereum blockchain. Höglund et al. [57] propose in their publication from 2023 a lightweight alternative to OCSP, namely TinyOCSP, and a protocol for the distribution of compressed certificate revocation (CCRL) using bloom filters. Similarly, Duan et al. [42] also propose a new form of CRLs using bloom filters and Merkle hash trees to find a balance between the consumed bandwidth to distribute the CRL, the RAM usage by the device, and the level of security. Boudagidgue et al. [47] address in their publication from 2020 a cluster-based revocation mechanism. Cluster leaders (CL) observe the behavior of member nodes (MN). In the event that member nodes perform malicious actions, a CL revokes the MNs certificate. However, the distribution of revocation information as such is not discussed. Additionally, in Boudagidgue et al.'s paper [59] from 2023, the usage of short-lived certificates as a mechanism to avoid revocation is proposed.

In OPC UA, trust lists that store trusted CA certificates also incorporate CRLs. An OPC UA application should be able to handle CRLs. Moreover, *TrustListValidationOptions* can be set to enable checking the revocation status online, leading to contacting an OCSP endpoint during the certificate validation process. For CIP Security, from the publicly available resources that we presented in Section 4.2, we have found no information regarding the supply of CIP-Security-enabled devices with revocation information. In PROFINET and

IEC/IEEE 60802, there is no standardized way specified to supply revocation information to system components.

Regarding IT-domain-based standards, RFC 5280 [12] specifies the profile for the X.509 v2 certificate revocation list (CRL) for use on the Internet. The revocation function specified herein comprises a CA that periodically issues a signed data structure called a certificate revocation list (CRL). This time-stamped list itemizes revoked certificates and is signed by a CA or CRL issuer. Each certificate is identified by its serial number. An advantage of this revocation method is that CRLs may be distributed by exactly the same means as certificates themselves. However, one limitation of the CRL revocation method is that the time granularity of revocation is limited to the CRL issue period. CMP [14] states that PKI management protocols must support the production of CRLs by allowing certified end entities to make requests for the revocation of certificates. In CMC [19] both the request to revoke a certificate and to request the retrieval of CRLs are specified as optional services. For TAMP [24], the cryptographic message syntax (CMS) can optionally be used to transport CRLs. The Online Certificate Status Protocol (OCSP) [15] defines a protocol to determine the current status of certificates without requiring CRLs. This enables the retrieval of timely information. An OCSP client issues a status request to an OCSP responder and suspends acceptance of the certificate in question until the responder provides the corresponding status. Additionally, RFC 6066 [77] specifies that constrained clients may wish to receive OCSP information directly from the server and optionally for multiple certificates [31]. In ACME [39], the protocol provides facilities to request the revocation of a certificate. In SCEP [17], clients may request a CRL by using the *GetCRL* function. SCEP does not specify a method to request certificate revocation; this must be done by contacting a CA using non-SCEP-defined mechanisms.

### 5.2.7. Summary

To conclude, certificate management does not only comprise the initial deployment of certificates but also their renewal, removal, and revocation. Moreover, we conclude that not only the management of certificates is important, but also the management of private keys, trust anchors, and revocation information. In the following, we will highlight the extent to which different stakeholders are involved in these functions and how these functions can be mapped to the life cycle of IA components. Furthermore, we conclude that all industrial communication standards have their own protocol-specific mechanisms for the initial equipment with trust anchors, private keys, and certificates, as well as their renewal, removal, and partially their revocation. These commonalities again give rise to the development of a unified generic model that describes certificate management functions in a protocol-independent manner. Lastly, we observe that the revocation of certificates is still unaddressed in three out of four industrial communication standards. Even though some research papers and also IT-domain-based standards address this topic, they are not incorporated into the standards.

### 5.3. Involvement of Different Stakeholders

While analyzing the selected approaches, we observed that they consider the participation of different stakeholders involved in the life cycle of IA components. In most cases, this is closely linked to the life cycle stages, which are addressed in Section 5.4.

Runde et al. [34], Hausmann et al. [30], and Fischer et al. [28] consider IA component manufacturers and operators as stakeholders that are involved in certificate management processes (see Figure 7). The approaches have the commonality of proposing the use of device manufacturer and operator PKIs. Every device manufacturer shall operate its own PKI to provide device certificates that can be verified by device operators. This allows the secure initial authentication before the subsequent configuration with certificates, trust anchors, and keys. Moreover, this approach provides protection against product piracy of automation devices. As a standard for secure initial device identifiers, the IEEE 802.1 AR [26] standard is addressed by Fischer et al. and Runde et al. In order to define and

enforce trust relations and access conditions according to the specific needs of a concrete system, the operator of the system should also operate a PKI that issues operator-specific certificates. Danilchenko et al. [45] describe a bootstrapping solution that also involves a component manufacturer as a possible operator of the intermediary entity, namely SPDS, for the security bootstrapping process. At manufacturing or commissioning time, a device must be configured to have a certificate to verify the SPDS authenticity. Additionally, the manufacturer must create and install a key and a certificate, signed by a CA recognized by the SPDS on the device. An operator of the device can create an account with the device vendor to configure a security configuration that can be retrieved by the device from the SPDS.

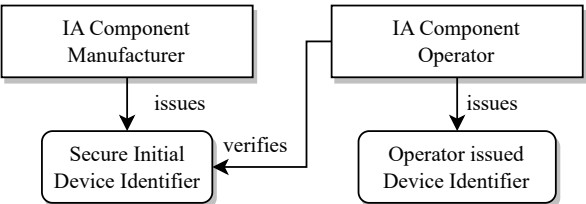

**Figure 7.** IA component manufacturer and operator certificate issuing domains.

In the current PROFINET specification, the usage of initial device identifiers (IDevIDs) according to IEEE 802.1 AR [26], allowing the secure identification and secure initial configuration of security parameters, is optional but recommended. We will elaborate more on the usage and issuance of certificates from the different domains in Section 5.4. In the current draft version of IEC/IEEE 60802 [60], manufacturers are even obliged to issue and deploy IDevIDs according to IEEE 802.1 AR [26] on their components. Similar to PROFINET, operators of devices must deploy their locally significant identifiers (LDevID) on their devices. The mechanisms to verify the IDevID of a device using, for example, PROFINET or IEC/IEEE 60802, are well-defined. However, the management procedure, how an operator receives a device manufacturer trust anchor that is used for the validation, is not specified.

In addition, CIP Security specifies, but does not require, the use of manufacturer-issued certificates. Interestingly, CIP Security also specifies the usage of self-signed certificates, generated during the manufacturing process or during the first power-up of the component while booting from the factory default state. The reason behind this certainly lies in the simplicity of this approach, because no manufacturer PKI needs to be created and maintained. However, a self-signed certificate can easily be spoofed, since an attacker might just generate himself a self-signed certificate [81]. In OPC UA Part 21 [56], the onboarding model of OPC UA devices is explained. This comprises a description of the stakeholders involved in the transfer of the physical device. The onboarding model addresses the involvement of *Manufacturers*, *CompositeBuilders*, *Distributors*, *Integrators*, and *OwnerOperators* of devices. Device *Manufacturers* create devices and issue *DeviceIdentity* certificates following the IEEE 802.1 AR [26] IDevID standard. *CompositeBuilders* build composites consisting of devices. *Distributors* resell devices or composites. In many cases, *Manufacturers*, *CompositeBuilders*, and *Distributors* belong to the same organizations. Similarly, *Integrators* that install and configure composites or devices and *OwnerOperators* that deploy and operate OPC UA systems often belong to the same organizations. When a transfer of physical control occurs, the supplier ships the device or composite with an electronic *ticket* that describes the equipment. A ticket is the term used for a document that describes one or more devices and has a digital signature that can be used to verify that the contents of the document have not been altered and that they confirm the origin of the device.

Regarding the approaches that originate mainly from the IT domain, BRSKI [35] requires that new unconfigured devices, so-called *pledges*, have an IDevID according to IEEE 802.1 AR [26] installed on them during the manufacturing process. The BRSKI

protocol requires a significant amount of communication between manufacturer and owner, providing a cryptographic transfer of control to the initial owner in its default mode. In its strongest mode, it identifies the owner in advance. The resale of devices is possible, provided that the manufacturer is willing to authorize the transfer. BRSKI enables the establishment of mutual trust between the *pledge* and the entity of the operator domain that discovers new pledges (called the *registrar*). A registrar authenticates ("Who is this device? What is its identity?") and authorizes ("Is it mine? Do I want it?") the pledge. At the same time, the pledge authenticates ("What is this registrar's identity?") and authorizes ("Should I join this network?") the registrar. BRSKI uses manufacturer-issued IDevIDs of the pledge to let the registrar authenticate and authorize it. It uses a new artifact called a "voucher" that it passes to the pledge to authenticate and authorize the registrar. This voucher is a cryptographically signed artifact signed by a manufacturer-authorized signing authority.

To conclude, we see that the approaches presented in research papers from Runde et al. [34], Hausmann et al. [30], and Fischer et al. [28] are reflected in a similar form in the industrial standards of PROFINET, IEC/IEEE 60802, CIP Security, and OPC UA. They all specify the usage of manufacturer-issued certificates for the authentication of devices and to secure the initial security configuration, even though not all of them require it. This requires manufacturers to operate PKIs and to make trust anchors accessible to owners or operators for the initial secure device identification and configuration. Moreover, all standards specify the use of operator-issued certificates. Interestingly, even though all industrial standards define a process that describes how an operator can authenticate devices added to the network, none of them define any mechanisms to allow devices to authenticate the network they are connected to. This implies that a device connected to a network allows itself to be configured via any network that it is connected to. All standards follow mechanisms based on "Trust on First Use" (TOFU) [82] or "Resurrecting Duckling" [83].

### 5.4. Life Cycle Stages

In this section, we focus on the life cycle stages considered by the reviewed approaches from Section 4. Please note that different approaches use different naming conventions. In the following, we strive to harmonize comparable stages, even though they are named differently. Moreover, we discuss the certificate-management-related processes in the different stages.

The different life cycle stages that we identified are the manufacturing, onboarding, operation, and decommissioning phases; optionally, after the decommissioning phase, an IA component can restart with the onboarding phase again (see Figure 8).

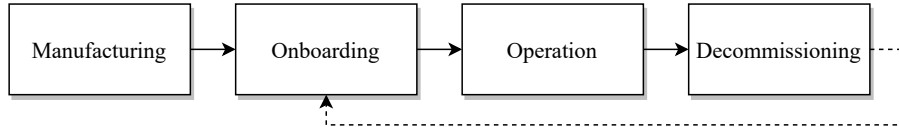

**Figure 8.** The approaches presented in Section 4 deal with certificate management in the life cycle stages of manufacturing, onboarding, operation, and decommissioning.

### 5.4.1. Manufacturing

During this initial manufacturing stage, a manufacturer creates an IA component. As already indicated in Section 5.3, Runde et al. [34], Hausmann et al. [30], Fischer et al. [28], and Danilchenko et al. [45], as well as industrial communication standards PROFINET, IEC/IEEE 60802, CIP Security, OPC UA, and the BRSKI [35] protocol describe the usage of manufacturer-issued and imprinted certificates. These certificates are used to authenticate the manufactured device. For this purpose, the party authenticating the devices needs to possess the trust anchor of the manufacturer's certificate. Once a device is manufactured, it is shipped to its owner or operator. OPC UA Part 21 [56] describes the life cycle of a device in an even more detailed manner, whereupon after manufacturing, the device is distributed

and potentially shipped to a *CompositeBuilder*. This collection of a singular device into a single unit is only addressed by OPC UA.

5.4.2. Onboarding, Bootstrapping, Commissioning, Initial Equipment with Credentials

As the heading already reveals, the subsequent life cycle stage is referred to by different names. They all have in common that (a) the device that has been shipped by the manufacturer or distributor is still in its factory default state and (b) after this stage, a device is initially equipped with cryptographic artifacts originating from the domain of the owner or operator.

As described in Section 5.3, Runde et al. [34], Hausmann et al. [30], and Fischer et al. [28] propose the usage of initial device identifiers as specified in IEEE 802.1 AR [26], to authenticate the devices during their initial configuration. During this initial configuration, an owner or operator of an automation system needs to issue and deploy certificates on the components from his own domain. This enables the authentication of devices and persons from his own domain. In particular, the configuration process is composed of a subgroup of the functions presented in Section 5.2. This includes the equipment with trust anchors, private keys, and certificates.

Additionally, Fischer et al. [28] note that authentication is the base for several security mechanisms, including authorization, integrity checks, and secure configuration. If the security level achieved during bootstrapping is already weak or vulnerable, other security measures that depend on them will be weak too. Moreover, the requirement is formulated that bootstrapping approaches shall minimize the management effort and therefore scale well with the number of devices. Similarly, Astorga et al. [11] present a commissioning phase, which is either the last phase of the manufacturing process or the first phase before the actual deployment of the device in its operational environment. In the commissioning phase, the new device is registered in the Certificate Life Cycle Management Server (CLM) and provided with an IBC key pair and networking configuration. In the following *enrollment* phase, the CLM can initiate the enrollment process by using a secured channel using DTLS and the raw public key from the IBC. In a similar manner, Höglund et al. [46] propose the usage of pre-loaded certificates by the manufacturer to identify the device and at least one CA certificate to identify the enrollment server of the operator. Then, in a fully automated manner, the device can contact the operator's CA using EST [27] to acquire a certificate, key, and trust anchor.

Danilchenko et al. [45] propose a commissioning phase after the manufacturing where an installer "claims" the devices and associates a previously configured security profile with the specific device. In Kohnhäuser et al.'s contribution [51], existing and emerging OPC UA provisioning solutions, including OPC UA Part 21, are analyzed and compared. According to the authors, OPC UA Part 21 assumes that a device is handled by a multitude of parties before it is put into operation, for example, a device is manufactured, distributed, assembled into a compound device by an integrator, and finally operated, maintained, and decommissioned. Therefore, in OPC UA Part 21, not only is the goal of providing secure device identification and authentication set, but also a secure log of all life cycle stages. This is done by introducing *tickets*, a separate digital document that describes a device and is signed by the party that handled the device. With this mechanism, each actor in the life cycle can use the *ticket* provided by the previous actor to establish trust in devices. Krishnan et al. [53] propose the usage of a provisioning server that is contacted and requested to issue a certificate. In contrast to the previous proposals, there is no initial authentication process described that verifies the device's identity. Garba et al.'s proposal [58] comprises a registration and configuration phase where a device registers its identity with a LRA and generates a self-signed certificate. According to the authors, the combination of the device's public key and the assigned identity is stored in the Ethereum blockchain and can be used to authenticate the device. Park et al. subdivide the onboarding phase into an *initialization* and *registration* stage prior to the usage of the secure MQTT-SN channel. In the initialization stage, device certificates are generated and installed on each device.

Moreover, in the registration stage, the broker is equipped with an access control list and the devices with a topic certificate.

There are also approaches that do not explicitly mention that there is such an onboarding phase and do not differentiate between different phases or stages of the life cycle of components [29,32,33,41,42,47,50,59]. However, Karthikeyan et al. [41] mention that implementing and maintaining a PKI in the domain of the owner or operator is challenging.

Regarding industrial communication standards, as described by Kohnhäuser et al. [51], the OPC foundation describes in its Part 21 [56] an onboarding model. In the onboarding stage, the system integrator connects a device to the network and verifies that the identity reported by the device matches the identity in the ticket provided by the manufacturer or composite builder. Every device shall have a device configuration application (DCA), which interacts with a registrar in the owner's or operator's network and performs the onboarding process. These exchanges between the device and the registrar are secured with the device identity certificate. After the authentication of the device, the DCA is issued a DCA certificate by the registrar. This DCA certificate allows all applications running on the device to automatically be onboarded and configured without human intervention. It shall not be used for communication with any application other than the registrar software update or certificate manager. As already explained, OPC UA Part 21 does not describe any mechanisms to allow devices to authenticate the network they are connected to, thus following the TOFU model. Therefore, in the onboarding stage, devices are exposed to malicious actors that have access to the network.

In CIP Security, the onboarding stage comprises the equipment of CIP-Security-enabled devices with certificates, private keys, and trust anchors issued by the domain of the owner or operator. When a device is in its factory default state, it shall either have a vendor-signed certificate following IEEE 802.1 AR [26] or a self-signed certificate. These default certificates will be stored in the CMO instance. Following the TOFU model, they can be used to supply the device with operator PKI-issued certificates, private keys, and trust anchors in a secured manner [81]. However, the vendor-signed certificate has the advantage that its authenticity can be verified, whereas the self-signed certificate can be spoofed by an attacker.

For PROFINET, similarly to OPC UA and CIP Security, the equipment or provisioning of PROFINET devices follows the TOFU model. Optionally, but not mandatory, devices can be equipped with manufacturer-issued certificates. They can be used to secure provisioning exchanges. Similarly to OPC UA, PROFINET distinguishes between two types of certificates that are provisioned: LDevID-Generic and LDevID-PN certificates. Firstly, LDevID-Generic certificates, corresponding trust anchors, and private keys are provisioned. This is done after the optional device authentication using the manufacturer-issued IDevID. The purpose of the LDevID-Generic is to protect the further management of the LDevID-PN, which contains information specific to the indented functionality and is used to protect the application data exchanges. Next to the management of LDevID-PNs, there are additional security configuration parameters managed by the SIH including for example the key usage threshold value after which symmetric keys need to be renewed.

In IEC/IEEE 60802, we also observe the application of the TOFU model and the usage of IDevIDs to authenticate devices prior to their equipment with owner- or operator-specific credentials and to secure these management exchanges. Moreover, this security setup sequence comprises the equipment with a trust anchor, an LDevID-NETCONF comprising a private key, an end-entity certificate, and a domain-specific certificate for name mapping. Thereupon, the IA station can use the LDevID-NETFCONF and the corresponding imprinted trust anchor to mutually authenticate NETCONF clients for secure configuration. The cert-to-name mapping maps the client certificate to a NETCONF username and is a prerequisite to checking the NETCONF client authorization.

Regarding standards originating from the IT domain, CMP [14] describes the initial registration and certification when an end entity requests a first certificate from a CA. The IETF RFC [14] only describes as a precondition that the end entity can authenticate the

CA's signature based on out-of-band means and that the end entity and the CA share a symmetric MACing key. However, it does not describe how these preconditions can be met. The CMC [19] and the draft standard of EALS [40] allow the usage of pre-shared-secrets to authenticate the exchanges between a CA and an EE. However, both approaches do not describe how to supply this pre-shared-secret. Regarding the EST protocol [27], an EST server authenticates and authorizes an EST client either using TLS with a previously issued client certificate issued by the EST CA or a manufacturer-installed certificate, certificate-less TLS (e.g., with a shared secret that is distributed out-of-band), or HTTP-based with a username/password distributed out-of-band. Moreover, clients need to authenticate EST servers. Therefore, the RFC [27] either assumes that EST clients possess a trust anchor to validate the EST server certificate or can be bootstrapped with this trust anchor. In AMCE [39], an ACME client contacts a CA and requests a certificate for an intended domain name. The contacted CA verifies that the client controls the requested domain name by having the client perform some action that can only be performed with control of the domain. Once the CA is satisfied, it issues a certificate that can be downloaded by the ACME client. The SCEP RFC [17] mentions that a client must retrieve CA certificates before any operation can happen. This CA certificate may be provided out of band, or a fingerprint of the certificate may be used to authenticate a CA certificate. Moreover, client authentication must be achieved by the client using an appropriate client certificate, which is either self-signed or issued by the SCEP CA or an alternate CA. In order to achieve enrollment authorization, a *challengePassword* attribute shall be sent as part of the enrollment request.

As the name indicates, the BRSKI RFC [35] specifically focuses on the bootstrapping process. In this process, an IEEE 802.1AR [26] IDevID is used within TLS to authenticate the device and to make a decision if one wants to take the device into possession. Moreover, the device uses a voucher artifact in order to authenticate the identity of the new owner or operator registrar and to decide whether to join the owner or operator network. As a result of the bootstrapping process, the device (pledge) stores a root certificate sufficient for verifying the registrar identity that enables the establishment of a TLS connection to an EST server to be enrolled with owner- or operator-specific certificates.

To conclude, all industrial communication standards pursue the usage of IEEE 802.1AR [26] IDevIDs to validate the authenticity of devices prior to their equipment with owner or operator domain-specific trust anchors, private keys, and trust anchors. Moreover, they are used to protect these initial equipment exchanges. Only OPC UA enables authentication of the origin of all prior participants in the supply chain using tickets, whereas other standards only enable authentication of the manufacturing domain. Furthermore, all industrial communication standards rely on the TOFU model. In comparison, regarding IT-domain-based standards, only BRSKI enables a mutually authenticated and authorized decision whether to equip devices with certificates or not. However, most of the other IETF RFCs that are presented in Section 4.3 do not explicitly detail how the artifacts used to secure the initial equipment (certificates or pre-shared keys) are supplied. Hausmann et al. [30], Runde et al. [34], Fischer et al. [28], and Höglund et al. [46] describe quite similar approaches for the initial equipment of devices compared to the existing industrial communication standards.

### 5.4.3. Operational

Once the initial equipment of IA components with certificates, private keys, and trust anchors is completed, an IA component can pursue its tasks, potentially through one or more applications running on the device. Throughout this subsequent operational stage, it may occur that a renewal of a certificate with trust anchor and/or private key renewal is necessary.

Regarding the research papers presented in Section 4.1, some only address the initial equipment but do not address the management during the operational phase [28,41,45,51]. Fernbach et al. [29] mention that due to their limited validity period, certificates need to be renewed in case they are expired. Krishnan et al. [53] also mention that in the

scenario where a particular device's certificate validity is over, it needs to be renewed, which is handled by the bootstrapping service they proposed. Astorga et al. [11] deal with the operational phase with their proposed CLM that checks renewal dates and starts re-enrolment when needed using SCEP [17]. Garba et al. [58] describe the usage of the certificate after the equipment as well as the update and revocation that can occur in the certificate life cycle. Park et al. [48] propose a dedicated re-keying protocol in case the topic membership changes during the operational stage. Höglund et al. [46] describe an overview of the life cycle of an IoT device where, after the bootstrapping, the device is in an operational state, is enrolled, and requires re-enrollment in order to avoid the expiration of the device's certificate.

OPC UA Part 21 [56] describes the operational stage as all applications on the device running normally and performing their intended tasks. Moreover, it shall be possible at this stage to update the trust list and/or renew the application instance certificate. Furthermore, a device can be reconfigured when applications are newly installed, modified, backed up, or restored. In this configuration state, a new device can be dropped in as a replacement for a to-be-exchanged device. Then, the replacement device can use the DCA certificate that was deployed during the onboarding procedure to automatically request additional certificates and trust lists without the need for additional approvals. PROFINET [52] supports a similar feature that significantly shapes PROFINET's specified certificate management life cycle by enabling the device replacement in an automated manner. Since the LDevID-Generic certificate allows the unattended management of LDevID-PN certificates, the replacement of a device is heavily simplified. The initial configuration with the LDevID-Generic is separated in a timely manner from the automated imprinting of the LDevID-PN. Moreover, this also simplifies management, including the renewal of certificates for devices that are already in the operational stage. IEC/IEEE 60802 also follows the same approach and uses the LDevID-NETCONF to enable the automated and unattended management of further certificates, keys, and trust anchors. Similarly, CIP Security supports a feature, namely Automatic Policy Deployment, that leverages the introduced pull model to initiate the deployment of security policies including a signed certificate from a CA [84].

In IT-domain-based standards, we do not face differentiation in an actual operational phase. However, as we discussed in Section 5.2, there are some standards, including CMC [19], TAMP [24], EST [27], ACME [39], and SCEP [17], that feature an explicit function to renew certificates.

To conclude, we observe that throughout the operational phase, a renewal of certificates or reconfiguration is necessary. Moreover, the replacement of devices can occur without prior notice. We observe that OPC UA [56], PROFINET [52], and IEC/IEEE 60802 [60] use a dedicated credential that enables automated configuration and equipment with additional credentials without further attendance.

### 5.4.4. Decommissioning

At the end of the operational phase, an IA component is disposed, discarded, or re-onboarded, for example, by re-selling to another owner. The only research paper that mentions this aspect is Höglund et al.'s [46] publication from 2020. The authors note that overwriting pre-installed CA certificates that were part of the enrollment process is beneficial. However, this introduces new complexity if the node is resold and needs to be reset to factory defaults.

Regarding industrial communication standards, OPC UA explicitly mentions the decommissioning stage in their onboarding model of Part 21 [56]. In the decommissioning stage, all access has to be revoked, and if the device is still functional, it is reset to the default factory settings. As explained in Section 5.2, a device has to proceed with the onboarding procedure from the beginning. Even though it is not explicitly mentioned in the PROFINET specification, we expect that with the resetting to the factory default settings, the imprinted certificates will also be removed, and the PROFINET component will again be in a state

where the onboarding process can be restarted. Similarly, we expect this for IEC/IEEE 60802 and CIP Security.

Regarding IT-domain-based standards, as we already explained in Section 5.2, BRSKI specifies the reset to the factory default state in two ways: a hard reset that removes all certificates, private keys, and trust anchors, and a soft reset that does not remove the trust anchors.

## 6. Research Gaps

In this section, we summarize the research gaps that we identified in the previous sections that are to be filled in the field of life-cycle-oriented certificate management in industrial networking environments.

First, considering the approaches that we presented in the industrial communication standards in Section 4.2, we observe that even though every approach has its own protocol-specific mechanisms, there are indeed similarities regarding their end entity reference architectures, supported functions, and life cycle stages. Especially, the manufacturing and initial equipment are very similar. Therefore, we believe that an overarching technology-independent generic concept that describes the management of certificates throughout the whole life cycle of IA components is needed. We believe that the benefits would be twofold. On the one hand, this would help develop a harmonized understanding of how certificates can be managed in industrial networking environments that allows for contextualizing existing certificate management approaches. The goal would be to map existing approaches into this technology- and protocol-independent generic concept. On the other hand, this base concept could be applied to other technologies that currently do not specify the management of certificates. Thus, it can serve as a generic blueprint for future technologies that currently do not support certificate management and therefore facilitate the large-scale deployment of certificate-based authentication mechanisms.

Second, to clarify the applicability of the aforementioned generic model, we believe that a concrete technology mapping to an industrial communication protocol that does not yet support certificate management should be elaborated. Technologies belonging to group (2) or even (1) from Section 4.2 are especially in scope to be extended with certificate management capabilities. This will again prove the feasibility of the concept and lead to the large-scale deployment and usage of certificate-based authentication mechanisms.

Third, especially when considering the functions of industrial communication standards (see Table 6), it becomes obvious that the question of how to address certificate revocation in real-world protocols is still unsolved, as any of the well-established mechanisms from the IT domain require either correct timing information, considerable memory, or an always online responder entity. This is also highlighted by Hausmann et al. [30], who mention that a design challenge related to the usage of a PKI for automation systems is to find a suitable solution for the revocation of certificates. Even though there exist different mechanisms proposed in IT-domain-based standards (e.g., CRLs, OCSP or OCSP stapling) and proposals from academic works (e.g., TinyOCSP [57], compressed CRLs [57], or flexible CRLs [42]), none of these are reflected in industrial networking standards. This indicates that the currently presented revocation mechanisms are not suitable for industrial communication standards.

Fourth, we believe that certificate management in industrial networking environments still requires a high amount of manual interaction and understanding of PKIs and their security principles, which do not necessarily belong to the core competencies of the owners and operators of IA components. Thus, we believe that owners and operators can be easily overwhelmed by the task of setting up and maintaining a PKI. Therefore, we believe that some guidance is required to support owners and operators. This is emphasized by Hausmann et al. [30] by stating that PKI configuration applications should hide related technical details as much as possible and support the creation, administration, and distribution of keys and certificates. In addition, Karthikeyan et al. [41] mention that implementing and maintaining a PKI in the domain of the owner or operator is challenging. Currently, specifi-

cations describe the management of certificates. However, the actual implementation and continuous maintenance throughout the life cycle of IA components need to be understood and executed by owners and operators who may have never read the specification.

## 7. Conclusions

This section finally concludes the work at hand. To this end, we refer to the research questions that we formulated in Section 3. In order to address **RQ1**, we followed a methodological-formal approach, which we explained in detail in Section 3. With the selected search query strings, we conducted searches in the databases IEEE Xplore, Science Direct, Scopus, and Springer Link. We iteratively condensed the number of stored results from 2042 to 20 results that we regard as relevant literature in the research field. Moreover, we searched for industrial communication standards that address the topic of certificate management. Lastly, we also researched IT-domain-based standards dealing with certificate management. An overview, timeline, and information about which approach is mentioned, used, or referenced by which approach is provided in Figure 5. The timeline gave us the insight that while the topic of certificate management has been addressed by the IETF in the past, its integration into industrial communication standards came much later and is still subject to updates. Additionally, taking Figure 5 into account, the topic is still subject to ongoing research. Moreover, to the best of our knowledge, there exists no work yet that surveys the topic.

With the selection of approaches at hand, we addressed **RQ2** by summarizing their key proposals. For the research papers, we provided a summary of the findings in Table 4. For the IT-domain-based approaches, please refer to Table 5.

In order to not only present the approaches separately, we address **RQ3** by addressing in Section 5 overarching aspects that enable a synopsis of the approaches. Therefore, we presented different types of entity reference architectures, compared certificate management functions, and presented the involvement of different stakeholders and life cycle stages. We differentiated the life cycle of an IA component in its manufacturing, onboarding, operation, and decommissioning phases. When comparing the industrial communication standards, we found that despite the use of different protocols, the overarching, technology-independent approach for the initial provisioning of certificates, private keys, and trust anchors is very similar. This also applies to the reference architectures of the entities, the involvement of the participants, and the certificate management functions.

Finally, we addressed **RQ4** and derived in Section 6 four open research gaps, where we believe that there are to-be-filled spots in the landmark of certificate management in industrial networking environments. We plan to address the identified research gaps in future research work.

**Author Contributions:** Conceptualization, J.G.; methodology, J.G. and A.W.; validation, J.G., A.W. and A.S.; investigation, J.G. and A.W.; writing—original draft preparation, J.G.; writing—review and editing, J.G., A.W. and A.S.; visualization, J.G.; supervision, A.S. All authors have read and agreed to the published version of the manuscript.

**Funding:** This work has been funded by the AiF within the program for sponsorship by Industrial Joint Research (IGF) of the German Federal Ministry of Economic Affairs and Energy based on an enactment of the German Parliament as part of the "FieldPKI" project under the grant number AIF/DFAM21752N.

**Acknowledgments:** We acknowledge support by the Open Access Publication Fund of the Offenburg University of Applied Sciences.

**Conflicts of Interest:** The authors declare no conflicts of interest.

## Abbreviations

The following abbreviations are used in this manuscript:

| | |
|---|---|
| CA | Certificate authority |
| CMO | Certificate Management Object |
| CRL | Certificate revocation list |
| CSR | Certificate signing request |
| DCA | Device configuration application |
| EE | End entity |
| GDS | Global Discovery Server |
| IA | Industrial automation |
| ICS | Industrial control systems |
| IT | Information technology |
| OCSP | Online certificate status protocol |
| OT | Operational technology |
| PKI | Public key infrastructure |
| RA | Registration authority |
| SIH | Security Infrastructure Handler |
| SPDS | Security Profile Distribution Service |
| TDME | TSN domain management entity |
| TOFU | Trust on first use |
| TSN | Time-sensitive networking |
| VA | Validation authority |

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
