# Peer review of "A Survey on Life-Cycle-Oriented Certificate Management in Industrial Networking Environments"

_jsan, doi:10.3390/jsan13020026_

Round 1

Reviewer 1 Report

Comments and Suggestions for Authors

The paper is fine scientifically.

This paper is a survey on the topic of certificate management in industrial networking environments, which is an up-to-date area with research potential.

Overall the paper is well drafted, which provides a comprehensive literature summary in the related area. This kind of survey paper is rarely published in the area of certificate management in industrial networking, which, however, could give unique insight to be referred by the future work. The survey methodology is clearly provided, which is efficient to provide a convincing conclusion, based on the selected references.

My suggestion goes into the reference list. More referrable and representative work should be considered in the survey.

Author Response

  • Since also other reviewers asked for additional references regarding vulnerabilities of industrial networking environments and the related challenges, we added additional references covering these aspects. 

Reviewer 2 Report

Comments and Suggestions for Authors

The abstract needs to make clear what the findings of the research are and how the findings make a contribution to the field of study. It seems that some sort of analysis was undertaken but it is not clear how it has underpinned knowledge and our understanding of the subject. Are there any recommendations? If yes, they need to be made clear.

Page 1: Introduction  - would it be useful to outline the type of attacks being referred to? This would provide additional context.

Page 2: what is the theoretical contribution? What aspect of academic theory is this research contributing to? This needs to be explained.

Page 4: Should an emphasis be placed on academic theory also? It is not clear what the research to be addressed is. The questions cited are more like research objectives!

There are a number of security related issues and challenges, but these have not really been identified and addressed as such. Would it be useful to do this?

Would it also be useful to cite the vulnerabilities and how the approach put forward can eradicate such vulnerabilities? And then link with security theory as this would provide a wider coverage of the topic and place more emphasis on the management approach.

What are the points of concern vis-à-vis intra – and inter-organizational relationships and problems arising that need to be overcome? Some of the key concerns have been raised in the paper but they could be placed in a management context. For example, who takes responsibility? How are relationships maintained through time?

Author Response

  • We explicitly added the findings to the abstract. This comprises: 
    • there is no work yet that surveys the research area of certificate management in industrial communication environments;
    • even though the first contributions in the IT-domain were made significantly earlier, the contributions to industrial communication standards were made significantly later;
    • industrial communication standards show distinct similarities regarding their considerations of the life cycle stages, stakeholders, and mechanisms 
  • We believe that a systematic review of attacks and vulnerabilities is worth a publication on its own. We gave the reference to Hemsley et al. as an overview on the history on ICS cyber incidents. Additionally, we now added another reference that reviews cyber vulnerabilities of communication protocols in indsutrial control systems. However, we believe anyone interested in the topic of certificate management in industrial networking environments is already familiar with the necessity to manage certificates as a precondition to establish authenticated communication channels. 
  • We believe that the topic of certificate management rather arises from a practical problem than a topic from the field of academic theory. Therefore, our goal is not the development new academic theories, but rather the collection and validation of existing theories.
  • See answer above, the goal is to provide a survey that collects and validates existing research than contributing new academic theories.
  • See answer to 2nd comment: we referenced Hemsley et al. and Xu et al. as entry points for vulnerabilities and attacks. We believe that a systematic review of those would give enough material for a publication on its own. Since the manuscript already is lengthy, we did not go into more detail.
  • See answer above, we addressed challenges like the "automated replacement of a device" or the "secure initial onboarding" process as well as how the different approaches handled these challenges. 
  • We extended this section: this requires device manufacturers that issue, for example, initial device identifiers (IDevIDs) according to IEEE 802.1AR to operate and maintain a PKI and to make trust anchors accessible to owners or operators for the initial device identification. Interestingly, the industrial communication standards do not specify how manufacturer-issued trust anchors are distributed to the owners or operators. However, they describe the protocol-specific means how to validate IDevIDs (e.g., using EAP-TLS in case of PROFINET). Additionally, we elaborate on the usage of "OPC UA Tickets" that especially deal with challenges of verifying the origin of another stakeholder during the transfer of a physical device.

Reviewer 3 Report

Comments and Suggestions for Authors

The paper conducts a comprehensive survey on life cycle-oriented certificate management within industrial networking environments, underscored by the Industry 4.0 movement and the emerging convergence of Operational Technology (OT) and Information Technology (IT) networks. This convergence, while facilitating enhanced connectivity, also enlarges the attack surface of industrial networks, necessitating robust cybersecurity measures. A significant component of these measures is the use of certificate-based authentication for industrial components at the field level, which requires meticulous management of certificates, private keys, and trust anchors across the components' lifecycle. The survey meticulously sifts through 2,042 initial results from databases like IEEE Xplore and SpringerLink, eventually narrowing down to 20 significant contributions that span research papers, industrial standards, and IT domain standards. By comparing these approaches, the paper delineates various certificate management functions, the involvement of different stakeholders, and the lifecycle stages of industrial components, culminating in the identification of prevailing research gaps. However, despite its thorough approach and insightful findings, the paper could benefit from further enhancements to deepen its impact and utility.

•    Detailed Analysis of Lifecycle Management Functions (Section 5.2): The paper provides a broad overview of certificate management functions such as equipment with trust anchors, private keys, and certificates, along with renewal, removal, and revocation functions. Could you expand on the technical mechanisms underlying these functions, especially in the context of industrial environments? For instance, how do specific protocols or standards address the unique challenges of certificate renewal in industrial networks where device accessibility might be restricted?
•    Stakeholder Involvement and Responsibilities (Section 5.3): While the survey mentions different stakeholders involved in certificate management, a deeper exploration of their specific roles, responsibilities, and interactions would be valuable. How do these roles vary across different industrial settings, and what best practices can ensure effective stakeholder collaboration?
•    Comparative Analysis of Certificate Management Approaches (Section 5): The survey presents a comparison of certificate management approaches from research papers, industrial standards, and the IT domain. Can you provide a more detailed comparative analysis, focusing on the advantages, limitations, and suitability of these approaches for different industrial applications? For example, how do the scalability, security, and ease of deployment compare across these approaches?
•    Research Gaps and Future Directions (Section 6): The identification of research gaps is crucial for advancing the field of certificate management in industrial networks. Can the paper delve deeper into specific technological, security, and interoperability challenges not yet addressed by current approaches? Additionally, proposing a roadmap for future research, potentially highlighting emerging technologies like blockchain or quantum-resistant algorithms, could significantly enhance the paper's contribution.
•    Impact of Emerging Technologies (Discussion Section): The rapid evolution of technologies such as machine learning, blockchain, and quantum computing presents both opportunities and challenges for certificate management. How might these technologies be leveraged to improve certificate management in industrial environments, and what are the potential pitfalls to be wary of?

Comments on the Quality of English Language

Minor editing of English language required

Author Response

  • We extended the "renewal function" subchapter (5.2) by including that PROFINET, IEC/IEEE 60802, and OPC UA, specify that devices are provisioned with an additional dedicated certificate used for the mutual authentication with a management entity. This enables to establish a secured communication for the security configuration including the seamless renewal of certificates over network means. 
  • We extended this section (5.3) by elaborating on the usage of "OPC UA Tickets" that especially deal with the challenge of verifying the origin of another stakeholder during the transfer of a physical device. Additionally, we explicitly state that device manufacturers that issue, for example, initial device identifiers (IDevIDs) according to IEEE 802.1AR are required to operate and maintain a PKI and to make trust anchors accessible to owners or operators for the initial device identification. The industrial communication standards do not specify how manufacturer issued trust anchors are distributed to the owners or operators. However, they describe the protocol-specific means how to validate IDevIDs (e.g., using EAP-TLS in case of PROFINET). 
  • We referred to Fischer et al. and Kohnhäuser et al. who both assessed approaches taking criteria like the scalability, ease of use, and security into account. However, since we tried in a first step to survey existing literature and since our manuscript already is lengthy, we did not go into further detail. Especially the aspect of usability is mentioned in detail when it comes to the feature of automatic device replacement. Which is very similarly realized in all industrial communication standards. Moreover, we also compared the approach to internally generate a self-signed certificate (CIP Security) vs. the provisioning with CA signed certificates (PROFINET, OPC UA, IEC/IEEE 60802, and CIP Security).
  • We pointed out the identified research gaps and mentioned that we plan to fill those in future publications. The mentioned directions certainly are interesting research paths. However, we believe that a decent elaboration on them would certainly go beyond the scope of this manuscript and is rather suitable for a complete research project on its own.
  • See comment above.

Reviewer 4 Report

Comments and Suggestions for Authors

The paper surveys the life cycle-oriented certificate management for industrial environments. The paper is well-written and organized. However, some issues need to be solved:

1. The abstract needs to be more concise. There is a lot of information that can be dropped.

2. Table 1: the total number of papers for Query#3 is incorrect. Please check the values once again.

3. Table 3: the exclusion criteria #2 is confusing since all the bibliographic databases included in the paper are subscription-based.  What "further payment" means in the context?

4. The captures (i.e., titles) of Figures 6 and 8 are confusing. Please reshape.

5.  The sentence from lines 1045-1047 (the one presenting the second research gap) is confusing and needs to be reshaped.

6. The last section, is more like a summary of the paper, no conclusions being presented. 

7. English needs some polishing.

Comments on the Quality of English Language

English needs polishing.

Author Response

  • We left out details regarding the synopsis and, since other reviewers asked, added information regarding the conclusion and gained insights instead.
  • We checked the values once again and corrected the number. 
  • We removed it and referred only to the subscription-based databases that are available at our institution. 
  • We reshaped the captures.
  • We reshaped the sentence.
  • We added further insights to the conclusion.
  • We polished all the sections to be revised.

Round 2

Reviewer 3 Report

Comments and Suggestions for Authors

The author address all the previous comments but still some issues are pending.

•    The abstract needs to be rewritten to point out significance and impact of the paper.
•    In the related work, it is recommended to refer the contribution made by the researchers and the novelty of the research. However, the author does not mention that.
•    I recommend that the authors add some more current articles to improve the paper's overall quality. The preparation of a comparative analysis of the current publications on this subject should also be included.
•    Avoid presenting with lengthy paragraph.

Comments on the Quality of English Language

Minor editing of English language required

Author Response

  • Since our manuscript presents a survey paper, our goal is not to present new impactful theories, but rather to collect and validate existing theories. Therefore, in the abstract, we limit ourselves to giving a short overview of our methodology, results, and key insights. Moreover, we mention that, to the best of our knowledge, this is the first work that surveys the topic of life cycle oriented certificate management in industrial networking environments, thereby pointing out the significance of our work. 
  • We do not have a “related work” section. Since our manuscript presents a survey paper, we limit ourselves to only present related works and provide a synopsis of those. In our Section 4 “Presentation of Certificate Management Approaches”, we present the individual contributions of the approaches. Table 4 and 5 provide an overview of the contributions. 
  • Since also other reviewers asked for additional references regarding vulnerabilities of industrial networking environments and the related challenges, we added additional references covering these aspects. Regarding the selection of contributions that we considered, we explicitly explained our selection methodology.
  • We divided long paragraphs into smaller subsections to increase the overall readability.

Reviewer 4 Report

Comments and Suggestions for Authors

The paper was substantially improved. All my comments and concerns were solved.

Comments on the Quality of English Language

Minor polishing needed.

Author Response

  • We strived to increase the overall readability.